# Safe and Scalable Web Agent Learning via Recreated Websites

Hyungjoo Chae [1]   Jungsoo Park [1]   Alan Ritter [1]

## Abstract

Training autonomous web agents is fundamentally limited by the environments they learn from: real-world websites are unsafe to explore, hard to reset, and rarely provide verifiable feedback. We propose VERIENV, a framework that treats language models as environment creators, automatically cloning real-world websites into fully executable, verifiable synthetic environments. By exposing controlled internal access via a Python SDK, VERIENV enables agents to self-generate tasks with deterministic, programmatically verifiable rewards, eliminating reliance on heuristic or LLM-based judges. This design decouples agent learning from unsafe real-world interaction while enabling scalable self-evolution through environment expansion. Through experiments on web agent benchmarks, we show that agents trained with VERIENV generalize to unseen websites, achieve site-specific mastery through self-evolving training, and benefit from scaling the number of training environments. Code and resources will be released at https://github.com/kyle8581/VeriEnv upon acceptance.

## 1. Introduction

Autonomous computer agents that can proactively assist humans in real-world tasks are a central goal of artificial intelligence (Xie et al., 2024; Xu et al., 2024). Achieving this vision requires agents that can self-evolve: continuously generating new challenges, interacting with complex environments, and improving without relying on costly human data (Zhou et al., 2025b; Huang et al., 2025). Recent advances therefore explore reinforcement learning for web agents, where agents directly interact with real-world websites, autonomously create tasks, and learn through self-challenging paradigms (Qi et al., 2025). Because the web

[1]School of Interactive Computing, Georgia Institute of Technology. Correspondence to: Alan Ritter <alan.ritter@gatech.edu>.

*Proceedings of the 43rd International Conference on Machine Learning*, Seoul, South Korea. PMLR 306, 2026. Copyright 2026 by the author(s).

(a) Traditional Self-Evolution Paradigm

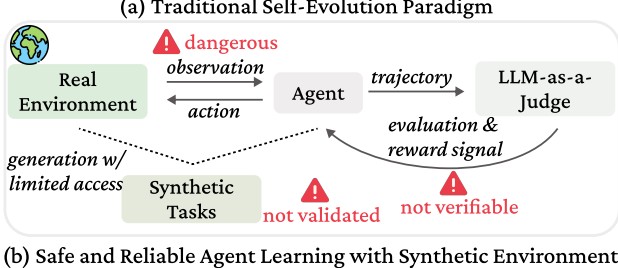

(b) Safe and Reliable Agent Learning with Synthetic Environment

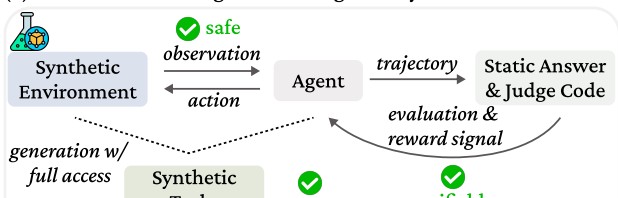

*Figure 1.* Comparison between the traditional self-evolution paradigm and our verifiable environment framework. (a) In traditional settings, agents interact directly with real-world environments and rely on unvalidated synthetic tasks and non-verifiable, LLM-based reward signals, leading to unsafe exploration and unreliable learning. (b) In contrast, VERIENV clones real-world websites into synthetic environments with full internal access, enabling safe exploration, validated task generation, and deterministic, verifiable reward signals for stable and scalable agent learning.

constitutes one of the most realistic and diverse computer-use environments, with long-horizon interactions, rich state, and heterogeneous interfaces (Zhou et al., 2024; He et al., 2024), it provides a natural testbed for scalable and general-purpose agent learning.

Despite their promise, learning directly from real-world websites introduces fundamental obstacles. First, such exploration is often **unsafe** or **restricted**: agent actions may interfere with other users, violate platform policies, or be blocked by mechanisms such as Cloudflare and CAPTCHAs. Second, self-generated tasks must be **well-specified**, **targeted**, and **executable**. Poorly specified or ill-defined tasks can misguide learning and invalidate reward signals. Prior work often generates underspecified instructions with multiple valid answers and relies on an LLM-as-a-judge to score trajectories (Zhou et al., 2025b). However, such LLM-based evaluation can be error-prone, whereas verification-based rewards are typically more reliable and robust (Garcia-Gasulla et al., 2025). Without reliable task definitions and verifiable

outcomes, self-evolving learning becomes unstable and inefficient. Consequently, effective self-evolving web agents critically depend on **both safe environments and verifiable task construction**.

We introduce VERIENV, a framework that automatically constructs safe, verifiable training environments for self-evolving web agents. As in Figure 1, rather than training agents directly on real-world websites, VERIENV uses a coding agent to automatically clone a target website into a fully executable synthetic environment, including its frontend, backend logic, and underlying database. This access allows tasks to be **generated alongside executable validation programs** (Zhou et al., 2025a; Wilf et al., 2025), enabling automatic validity checks and deterministic evaluation of agent trajectories. As a result, agents trained with VERIENV learn from **reliable, reproducible training signals** rather than heuristic or LLM-based judgments. By decoupling self-evolving learning from unsafe real-world exploration and grounding it in verifiable environments, VERIENV provides a practical and scalable foundation for training autonomous web agents.

In our experiments, we evaluate VERIENV from two complementary perspectives. First, using WebArena (Zhou et al., 2024) and Mind2Web-Online (Xue et al., 2025), we demonstrate that agents trained within our framework generalize to out-of-domain settings and realistic web tasks; on WebArena, VERIENV improves success rates by $+6.06$ (Qwen3-4B) and $+9.09$ (LLaMA-3.2-3B-Instruct) points over the corresponding base models. Second, we investigate whether an agent can achieve site-specific mastery through repeated training within a simulated environment cloned from a fixed website. Beyond these settings, we compare verifiable task generation against prior approaches (Zhou et al., 2025b), which generate tasks without direct environment access and rely on LLM-as-a-judge for trajectory evaluation. Our analysis highlights the importance of executable, verifiable tasks for stable agent learning and shows that agent performance improves as the number of training environments increases, indicating the effectiveness of environment scaling in self-evolving web agents.

Our contributions are summarized as follows:

- We propose VERIENV, a framework that automatically reconstructs real-world websites into executable synthetic environments and generates verifiable tasks, enabling safe and reliable self-evolving agent learning.

- Through extensive experiments on WebArena and Mind2Web-Online, we show that agents trained within VERIENV generalize effectively to unseen websites.

- We provide systematic analyses demonstrating the importance of verifiability in task construction and reward assignment, as well as the impact of environment scaling and coding agents on agent learning.

## 2. Related Work

**Agent learning with verifiable reward.** Learning agents for web interaction and tool use typically requires long-horizon trajectories with many sequential decisions, making learning signals sparse and brittle in unconstrained environments. Recent progress has therefore emphasized *verifiable* training signals and controlled settings where success can be evaluated reliably (Wilf et al., 2025). In math and coding, reinforcement learning with verifiable rewards improves reasoning and tool use by grounding learning in outcome-checkable feedback (Mai et al., 2025; Wen et al., 2025). Beyond single-shot problem solving, self-challenging setups further strengthen supervision by generating executable verifiers and tests (Zhou et al., 2025a). For web agents, structured pipelines that separate proposing, executing, and evaluating actions offer clearer reward semantics and more scalable skill acquisition (Zhou et al., 2025b). In contrast, VERIENV targets web settings where direct exploration is unsafe or blocked and outcomes are not externally verifiable, by cloning the full website (including its database) and enabling controlled internal validation for trajectory evaluation and reliable rewards.

**Self-evolving agents.** A complementary line of work studies how agents can *self-evolve* via exploration, curricula, and automated task construction, reducing reliance on static human supervision. Realistic benchmarks for web agents, such as Mind2Web (Deng et al., 2023), WebVoyager (He et al., 2024), and WebArena (Zhou et al., 2024), enable systematic study of end-to-end agents and iterative improvement. Building on these environments, methods increasingly use online curricula and self-evolving loops: WebRL adapts training tasks to target an agent's weaknesses over time (Qi et al., 2025), while other work scales coverage via exploration-driven task generation (Ramrakhya et al., 2025) or environment/task generation pipelines (Hu et al., 2025).

Similar self-evolution ideas also appear in reasoning-centric agents: corpus-grounded self-play induces automatic curricula (Liu et al., 2025), and reinforced self-training iteratively improves models using self-generated data with reinforcement-style filtering (Gulcehre et al., 2023). Whereas prior web-agent methods often rely on real-site interaction or unverifiable task generation, VERIENV clones real sites into executable environments with database-backed verification, enabling valid self-generated tasks and fully verifiable rewards without impacting real users or platform constraints.

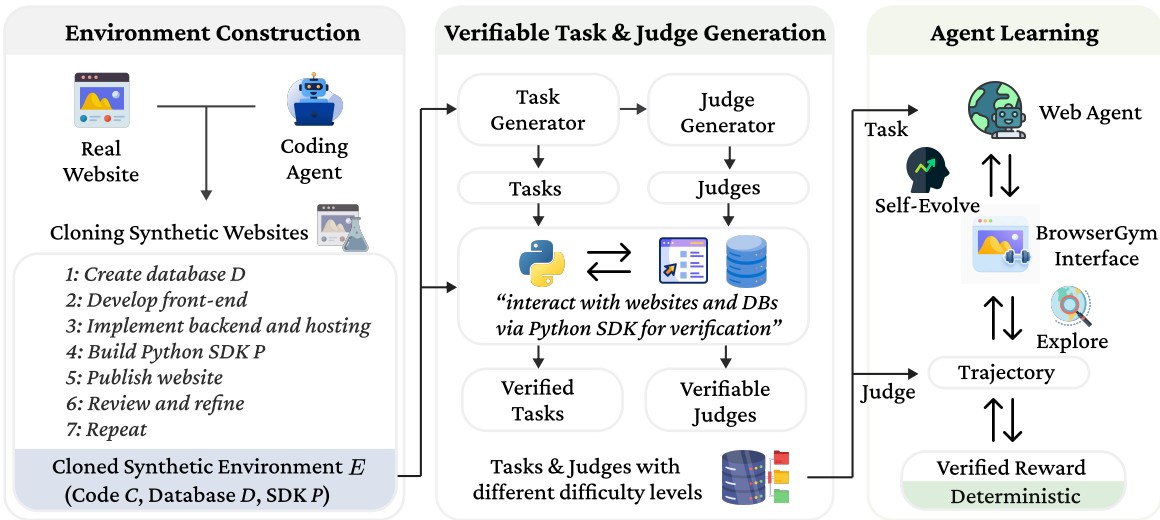

*Figure 2.* Overview of VERIENV. VERIENV first clones a real website into a fully instrumented synthetic environment (code $C$, database $D$, and a Python SDK $P$) via coding agent, then uses task and judge generators to produce tasks at varying difficulty and verify both tasks and judges by interacting with the website and database through the SDK, yielding deterministic, verified rewards for agent learning.

**Coding agents for web development.** Recent coding agents have demonstrated the ability to autonomously develop web applications end-to-end, ranging from frontend design and backend implementation to deployment (Yang et al., 2024; Jimenez et al., 2024), by leveraging tool calling for file system access, terminal execution, and external search (Wang et al., 2025). Despite their growing capabilities, such agents frequently introduce implementation errors and require iterative debugging (Chen et al., 2024), which they typically address by incorporating feedback from compiler outputs, runtime logs, language servers, and vision–language models (Muennighoff et al.; Chae et al., 2024; Zheng et al., 2024a). However, many critical bugs cannot be caught by static checks alone: functional failures, layout issues, and interaction errors often only appear during execution. Prior work therefore detects such bugs via website interaction using web agents and browser-based testing frameworks (Wang et al., 2025; Lu et al., 2025a;b). Building on this, we pair coding agents with automated web interaction to iteratively refine cloned sites, improving functionality and producing reliable synthetic environments.

## 3. Method

Our framework focuses on carefully preparing reliable environments where agents can safely train. We show the overall flow of our framework in Figure 2, where we (i) clone real-world websites into executable synthetic environments (Section 3.1), (ii) derive verifiable tasks and judges from these environments (Section 3.2), and (iii) train agents on the resulting tasks within the synthetic environments (Section 3.3).

### 3.1. Recreating Real-World Websites

We leverage a coding agent, GPT-5.2 (OpenAI, 2025), to construct a training environment that ensemble a real-world target website. Specifically, given screenshots of a real-world website $E$, a coding agent is tasked with reconstructing the service into a synthetic environment $\tilde{E}$. Toward that goal, the coding agent operates with local file system and terminal access, allowing it to freely write, execute, and iteratively refine code. Through this process, the agent produces an executable system that captures the core application logic and data semantics of the target service.

We represent the resulting synthetic environment $\tilde{E}$ as a tuple $(\mathcal{C}, \mathcal{D}, \mathcal{P})$, where $\mathcal{C}$ denotes the executable application code, $\mathcal{D}$ the underlying database state, and $\mathcal{P}$ a Python SDK that exposes controlled internal access for querying and verifying environment states. In addition to implementing the main application logic, the coding agent also creates auxiliary scripts for environment control, such as bash scripts for server startup and reset utilities, which facilitate repeated experimentation and agent training.

Because the reliability and interface complexity of websites are crucial for training agents, constructing high-quality synthetic environments requires substantial programming and debugging. Thus, after the initial implementation, the cloned environment is further refined through an iterative stabilization process. Imitating human developers' workflow (Lu et al., 2025a;b), the coding agent is encouraged to interact with the deployed website using Playwright MCP (Microsoft, 2024), identify functional discrepancies, and incrementally patch bugs based on observed failures. This iterative refinement results in a stable and resettable

synthetic environment suitable for reliable task execution, validation, and downstream agent learning. While the cloned environment is not perfectly identical to the original website, it preserves the functional structure necessary for verifiable and reproducible training.

### 3.2. Verifiable Task and Judge Generation

Given a synthetic environment $\tilde{E} = (\mathcal{C}, \mathcal{D}, \mathcal{P})$, we prompt large language models (LLMs) to generate tasks that can be automatically verified within the environment. Each task $\mathcal{T}$ is specified by a natural language description and a validation program using the Python SDK $\mathcal{P}$. The goal of this program is to (1) validate the executability of the generated task, and (2) create a verifiable judge. Each task includes a validation program, which specifies task success conditions using executable predicates over environment state. At the end of an episode, these predicates are instantiated as a verifiable judge, which deterministically evaluates the terminal state and returns a binary reward indicating task completion.

For example, in Figure 3, the task is to sort the list of apartments by price, and answer the name of the first item and its price. The following validation program first checks whether the task is valid by simulating the desired process, and returns the information to construct the verifiable judge (*e.g.*, `must_include("Reed-Hill Apartments")`). This process enables scalable task generation without manual annotation, while guaranteeing that task correctness can be deterministically assessed through executable verification rather than heuristic or LLM-based judgments. Figure 3 provides a concrete example of such a verifiable task, illustrating how natural language instructions are paired with executable validation programs. Such validation programs are subsequently used to compute deterministic reward signals during self-evolving agent learning, as described in the next section.

### 3.3. Self-Evolving Agent Learning in Verifiable Environments

Building on the automatically generated and verifiable tasks, agents are trained through a self-evolving learning loop within the synthetic environment $\tilde{E}$. At each iteration, an agent interacts with the cloned website to solve a sampled task $\mathcal{T}$, producing a trajectory $\tau$ consisting of browser actions and observations.

Upon task completion, the agent's trajectory $\tau$ is evaluated by executing the task-specific validation program through the Python SDK $\mathcal{P}$, which deterministically queries the underlying database state $\mathcal{D}$. This evaluation yields reproducible reward signals that are independent of heuristic or LLM-based judgments. The verified rewards are then used to update the agent, enabling stable and scalable learning without manual annotations or human supervision. We

**Task Instruction**

In Columbus, OH, sort listings by price (low to high). What is the first listing's name and minimum monthly price?

**Validation Program using Python SDK**

```python
from apartments_sdk import
ApartmentsClient
client = ApartmentsClient()
res = client.search_listings(q='Columbus,
OH', sort='price_asc', limit=1)
l = res.items[0]
result = {'id': l.id, 'name': l.name,
'min_price': l.min_price}
result
```

**Execution Result**

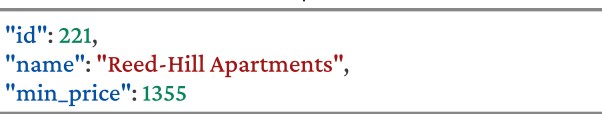

"id": 221,
"name": "Reed-Hill Apartments",
"min_price": 1355

**Verifiable Judge**

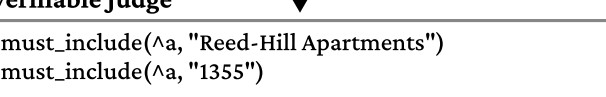

must_include(^a, "Reed-Hill Apartments")
must_include(^a, "1355")

*Figure 3.* Example of a verifiable task with executable validation in a synthetic website (*i.e.*, cloned from apartments.com).

*Table 1.* Statistics of constructed synthetic environments and generated tasks.

| Statistic | Value |
|---|---|
| Number of websites | 149 |
| Number of tasks per website | 49.5 |
| Total number of tasks | 7,400 |
| Easy tasks | 2,972 (40.2%) |
| Medium tasks | 2,900 (39.2%) |
| Hard tasks | 1,528 (20.6%) |

choose reward-based rejection fine-tuning as an example of a possible training method for utilizing the verifiable rewards. To support continual self-improvement, newly generated tasks and collected trajectories are iteratively incorporated into the training process. This self-evolving procedure allows agents to progressively adapt to increasingly complex behaviors while remaining grounded in verifiable environment feedback.

### 3.4. Environment Statistics and Human Evaluation

We construct synthetic environments for 149 websites, selected by referencing the website list used in Mind2Web (Deng et al., 2023) and Mind2Web-Online (Xue et al., 2025) to ensure coverage of realistic and diverse web domains. For each website, we generate 50 task instructions

*Table 2.* Comparison with existing web agent datasets and benchmarks. VERIENV uniquely enables verifiable evaluation and scalable task generation through executable synthetic environments.

| Dataset / Benchmark | # Websites | # Tasks | Browser Interaction | Verifiable Judge | Scalable Task Gen. |
|---|---|---|---|---|---|
| WebArena | 5 | 812 | ✓ | ✓ | ✗ |
| WorkArena | 1 | 33 | ✓ | ✓ | ✗ |
| WebVoyager | 15 | 643 | ✓ | ✗ | ✗ |
| Mind2Web | 137 | 2,350 | ✗ | ✓ | ✗ |
| Mind2Web-Online | 136 | 300 | ✓ | ✗ | ✗ |
| Mind2Web-Live | 137 | 542 | ✓ | ✗ | ✗ |
| **VeriEnv (Ours)** | **149** | **7,400** | ✓ (w/ synthetic websites) | ✓ | ✓ |

*Table 3.* Human evaluation of the generated websites and tasks.

| Metric | Result |
|---|---|
| **Environment quality** | |
| Functional correctness (avg.) | 90% |
| Signup | 94% |
| Login | 95% |
| Search | 81% |
| Filter | 88% |
| Navigation | 100% |
| Forms | 100% |
| Visual rating (Likert, 1–5) | 4.7 |
| **Task validity** | |
| Task executability | 90% |
| Judge correctness | 76% |

using large language models, resulting in a total of 7,400 tasks. Each task is annotated with a difficulty label (**easy**, **medium**, **hard**) based on predefined criteria reflecting action length, statefulness, and authentication requirements. Table 1 summarizes the statistics of the constructed environments and generated tasks. Also, in Table 2, we compare our environments with existing datasets and benchmarks. With our website recreation pipeline, we provide more diverse websites, while the instructions are verifiable.

To conduct a human evaluation, we recruited four graduate students with computer science background. Two annotators evaluate 15 instances each in one subset and two evaluate 15 instances each in a second subset, yielding double annotation per subset. Annotators rate environment quality via **Functionality** (success rate over signup, login, search, filter, navigation, and forms) and **Visual rating** (5-point Likert, higher is better). They also assess **task validity** with binary judgments of whether a task is executable on the synthetic website as described (**Task executability**) and whether the automated validator correctly determines task completion (**Judge correctness**).

Table 3 summarizes the results: functionality averages 90.3% success across capabilities and visual quality is rated 4.7/5, indicating high-quality synthetic websites. Task executability and judge correctness are 90% and 76%, respectively. The most common errors in judge correctness arise from database resets that do not preserve the random seeds used for populating website data. We find that such errors can be reliably detected and resolved by re-running the validation programs implemented with the Python SDK. Interannotator agreement on the binary judgments is substantial, with mean Cohen's $\kappa = 0.61$ (Cohen, 1960). Although judge correctness is lower than task executability, it remains informative because our validators are fully verifiable and rule-based.

Unlike model-based evaluators (*e.g.*, LLM-as-a-Judge (Xue et al., 2025)) that can introduce additional uncertainty in complex web environments, these checks yield deterministic, auditable pass/fail decisions when applicable, providing a conservative but reliable foundation for evaluation.

# 4. Experiments

This section evaluates VERIENV in two complementary settings. First, we study **cross-domain generalization** by training on recreated websites and testing on established benchmarks that cover unseen sites and tasks. Second, we study **site-specific mastery** by repeatedly training and self-evolving an agent within a single recreated website to measure in-domain improvements over time.

## 4.1. Generalization Across Websites

**Implementation details.** We implement VERIENV using GPT-5.2 (OpenAI, 2025) as the backbone LLM and Cursor CLI (Cursor, 2025) as the coding agent for environment construction. The cloning process takes 83.5 minutes and costs $3.6 per website on average, including the debugging and task generation process. The list of target websites and the screenshot of the websites are obtained from Mind2Web (Deng et al., 2023). To evaluate cross-domain generalization, we explicitly exclude websites that overlap with the test split of the evaluation benchmarks from the cloning and training process.

After constructing synthetic environments and generating verifiable tasks, we train web agents based on two open-

*Table 4.* WebArena-Lite evaluation results across different websites. Methods annotated with * use numbers reported by Chae et al. (2025).

| Method | Shopping | CMS | Reddit | GitLab | Map | Total | Δ |
|---|---|---|---|---|---|---|---|
| GPT-4o-mini (Hurst et al., 2024)* | 21.74 | 22.86 | 19.05 | 34.38 | 19.35 | 23.64 | – |
| GPT-4o (Hurst et al., 2024)* | 23.91 | 31.43 | 28.57 | 56.25 | 19.35 | 31.52 | – |
| Qwen3-4B | 3.77 | 6.67 | 4.17 | 13.89 | 14.29 | 7.88 | – |
| +Synatra (Ou et al., 2024) | 0.00 | 0.00 | 12.50 | 8.33 | 0.00 | 3.64 | −4.24 |
| +ADP (Song et al., 2025) | 4.35 | 5.71 | 9.52 | 3.13 | 9.68 | 6.06 | −1.82 |
| +VERIENV (Ours) | 4.35 | 20.00 | 23.81 | 12.50 | 16.13 | 13.94 | **+6.06** |
| LLaMA-3.2-3B-Instruct | 0.00 | 2.86 | 9.52 | 3.13 | 3.23 | 3.03 | – |
| +Synatra (Ou et al., 2024) | 2.17 | 2.86 | 14.29 | 9.38 | 6.45 | 6.06 | +3.03 |
| +ADP (Song et al., 2025) | 4.35 | 11.43 | 14.29 | 12.50 | 6.45 | 9.09 | +6.06 |
| +VERIENV (Ours) | 4.35 | 17.14 | 19.05 | 15.63 | 12.90 | 12.73 | **+9.70** |

*Table 5.* Mind2Web-Online results across difficulty levels. Methods annotated with * use numbers reported by Xue et al. (2025).

| Method | Easy | Medium | Hard | Total | Δ |
|---|---|---|---|---|---|
| Browser-Use-GPT-4o (Browser-Use Contributors, 2024)* | 55.40 | 26.60 | 8.10 | 30.00 | – |
| Claude-3.5-Sonnet (Anthropic, 2025)* | 56.60 | 26.60 | 6.80 | 28.80 | – |
| Qwen3-4B | 26.32 | 9.41 | 11.63 | 13.18 | – |
| +Synatra (Ou et al., 2024) | 35.09 | 5.88 | 9.30 | 14.55 | +1.37 |
| +ADP (Song et al., 2025) | 26.32 | 7.06 | 6.98 | 11.36 | −1.82 |
| +VERIENV (Ours) | 29.82 | 23.53 | 6.98 | 20.45 | **+7.27** |
| LLaMA-3.2-3B-Instruct | 19.30 | 12.94 | 0.00 | 11.36 | – |
| +Synatra (Ou et al., 2024) | 24.56 | 15.29 | 6.98 | 14.55 | +3.19 |
| +ADP (Song et al., 2025) | 42.11 | 24.71 | 11.63 | 24.09 | +12.73 |
| +VERIENV (Ours) | 40.35 | 29.41 | 13.95 | 24.55 | **+13.19** |

source base models: Qwen3-4B (Yang et al., 2025) and LLaMA-3.2-3B-Instruct (Dubey et al., 2024). To construct training data, we employ a rejection-based fine-tuning strategy on 97 websites. Specifically, for each generated task, we sample agent trajectories and retain only those that successfully satisfy the corresponding executable validation criteria. The resulting filtered trajectories are then used as supervised training data for agent fine-tuning, enabling stable learning from verifiable task completion signals. Additional implementation details, including training hyperparameters and system configurations, are provided in Appendix A.1.2.

**Benchmarks and baselines.** We evaluate agent performance on two widely used benchmarks for web agents: (1) WebArena-Lite (Zhou et al., 2024) measures task success across 5 realistic websites implemented within Docker. (2) Mind2Web-Online (Xue et al., 2025) focuses on generalization over 100+ real-world websites and provides 300 tasks with three difficulty levels– easy, medium, and hard. As some of the websites in Mind2Web-Online block web agents, we exclude tasks that have such issues, resulting in 220 tasks. We use the WebJudge-7B model from the original paper for trajectory evaluation.

We consider two categories of baselines. (1) **Proprietary LLMs**: GPT-4o-mini, GPT-4o (Hurst et al., 2024) and Claude-3.5-Sonnet (Anthropic, 2025), representing state-of-the-art closed-source models. (2) **Open-source web agents**:

models trained using existing web-agent datasets and training protocols. In particular, Synatra (Ou et al., 2024) constructs synthetic trajectories from website-specific tutorials, while Agent Data Protocol (ADP; Song et al. (2025)) aggregates multiple agent datasets and standardizes action representations. ADP aggregates diverse web-agent training datasets and provides them in a unified format, simplifying the training process.

**Result.** We show the results in WebArena and Mind2Web-Online on Table 4 and Table 5, respectively. ADP exhibits notably different behaviors depending on the base model. With LLaMA-3.2-3B-Instruct, ADP leads to a clear performance gain, particularly on Mind2Web-Online, which we attribute to dataset overlap between ADP's aggregated training data (including Mind2Web and related web interaction datasets such as NNetNav (Murty et al., 2025)) and the evaluation distribution. In contrast, ADP does not consistently benefit Qwen-based models and can even degrade performance. We observe that mixing heterogeneous datasets in ADP introduces issues in generating coherent reasoning and adhering to expected action formats, suggesting a mismatch between ADP's supervision signals and Qwen's action-generation behavior.

In contrast, VERIENV consistently improves performance across base models in the fully out-of-domain WebArena setting. We attribute this improvement to training on self-

generated trajectories with verified task completion, where successful runs provide structured thought and action tokens as implicit supervision signals. This self-evolving training paradigm encourages more stable learning of reasoning and action token distributions, resulting in improved generalization across both Qwen3-4B and LLaMA-3.2-3B-Instruct, with gains of $+6.06$ and $+9.09$ points, respectively.

### 4.2. Site-Specific Mastery via Self-Evolving Training

**Setup.** One compelling use case of VERIENV is site-specific mastery, where an agent is trained to excel on a particular website through repeated interaction. In this setting, a target website is cloned into a synthetic environment that serves as an effectively unbounded training gym for self-evolving agents. To study this scenario, we construct synthetic environments for websites drawn from WebArena and train web agents entirely within the cloned environments. Although WebArena provides sandboxed websites in which agent exploration is inherently safe, we treat these websites as proxies for real-world services. During training, agents are restricted to interact only with the cloned environments rather than the original WebArena instances. The goal of this experiment is to evaluate whether self-evolving training in a verifiable synthetic environment can lead to strong in-domain performance on a fixed website.

We compare VERIENV against PAE (Zhou et al., 2025b), a recent approach that generates tasks and uses vision language models for evaluating the trajectories. While both methods leverage automatically generated tasks for agent training, they differ fundamentally in their learning setup. PAE relies on real websites, non-verifiable tasks, and LLM-based judges for reward assignment, whereas VERIENV operates exclusively in synthetic environments and uses verifiable judges to provide deterministic reward signals.

**Result.** Figure 4 presents the results of site-specific self-evolving training across three representative website categories. Across all settings, agents trained with VERIENV consistently improve their performance as training progresses from the base model to later self-evolution phases, indicating that repeated interaction within a fixed, cloned environment effectively strengthens in-domain capabilities. VERIENV yields larger and more stable performance gains than PAE across training phases, with the strongest improvements in CMS and Shopping. While PAE benefits from iterative task generation, its non-verifiable tasks and LLM-judge evaluation constrain progress. VERIENV, by contrast, continues to improve throughout training, consistent with executable, verifiable rewards providing a more reliable learning signal.

These results indicate that verifiable synthetic environments are well-suited for site-specific mastery, enabling agents to

progressively refine their behaviors without requiring direct interaction with real-world websites. Unlike robotics domains, where sim-to-real gaps often pose a fundamental challenge (Peng et al., 2017), web environments exhibit a much smaller discrepancy between synthetic and real executions when the underlying functionality and state transitions are faithfully reproduced.

## 5. Analyses

This section provides additional analyses to clarify when and why VERIENV works. We study the scaling behavior as the number of recreated websites increases and analyze common failure modes in automated website construction.

### 5.1. Environment Scaling Effects on Web Agents

VERIENV is a fully automatic framework that enables scaling the number of training environments to broaden the coverage of web agents. We analyze how increasing the number of training environments influences agent performance by varying the portion of environments used during training and evaluating intermediate checkpoints on WebArena and Mind2Web-Online.

As shown in Figure 5, agent performance generally improves as the number of training environments increases across both benchmarks. The improvements follow a consistent upward trend within the evaluated range, indicating that additional environments provide useful learning signals for web agents. In contrast, baseline methods that rely on fixed datasets or non-verifiable supervision exhibit relatively stable performance, suggesting limited sensitivity to environment scaling. Overall, these results suggest that expanding the diversity of training environments can be beneficial for improving web agent capabilities under verifiable training settings. By enabling safe and systematic environment expansion, VERIENV facilitates a scalable training paradigm that complements existing approaches focused on data or model scaling.

### 5.2. Error Analysis on Environment Construction

To better understand the limitations of automated environment construction, we analyze the 39 websites (out of 136) that failed to be successfully implemented by our framework. Figure 7 summarizes the primary failure modes observed during the implementation and debugging process. The most common issues arise from incomplete system setup, such as missing server startup scripts and failed task generation, indicating that end-to-end orchestration remains a major challenge for coding agents. Among websites that reached a runnable state but still exhibited errors, infrastructure-related issues such as port conflicts and CORS misconfigurations account for the majority of failures. In particular,

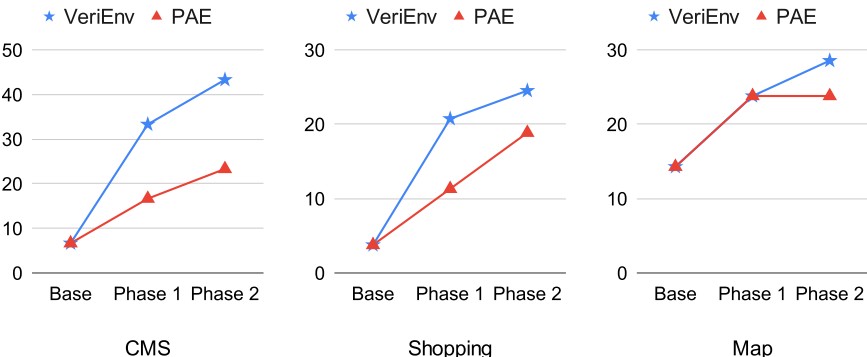

*Figure 4.* Site-specific self-evolving training within a cloned synthetic environment. Agents are trained on a fixed target website using automatically generated tasks and verifiable reward signals.

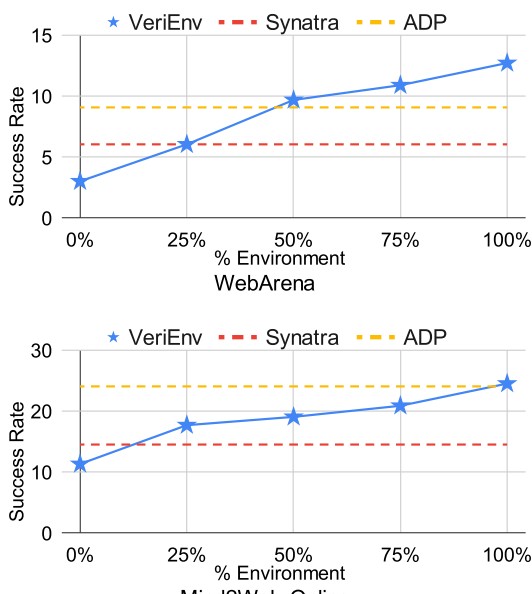

*Figure 5.* Analysis on the scaling effect of the number of websites.

port conflicts largely stem from deploying more than 100 web applications on a single server, and are not inherent limitations of the approach itself; with sufficient resources, isolating each website by using Docker would be a more reliable and scalable solution.

### 5.3. Comparison of VERIENV and PAE

Figure 6 compares PAE (Zhou et al., 2025b) with VERIENV. PAE generates tasks from real website interactions and tutorials, but some resulting tasks are ambiguous and admit multiple plausible answers. In such cases, the policy may fail to reach the intended target page (e.g., a recipe), yet a vision-language judge may still label the outcome as successful as long as it contains seemingly relevant information,

leading to false positives. In contrast, VERIENV constructs tasks with a single, well-defined answer and verifies the policy's terminal state using a rule-based checker backed by a Python SDK, enabling deterministic and reliable evaluation of trajectories.

## 6. Discussion and Future Directions

### 6.1. When does a coding agent struggle to recreate websites?

Although coding agents can recreate a wide range of websites, we observe several recurring scenarios where reconstruction quality degrades. In particular, websites that rely heavily on multimedia delivery are more challenging to reproduce faithfully. Platforms such as arXiv or YouTube involve serving PDF documents or video streams, which require additional infrastructure.

Importantly, these challenges do not fundamentally prevent environment reconstruction. In many cases, the functional behavior of the service can still be approximated by replacing such components with lightweight placeholders. For instance, coding agents can be instructed to serve dummy PDF files or sample video assets, enabling the reconstructed service to remain operational while avoiding the complexity of full media pipelines. Similarly, for image-intensive services such as shopping websites, realistic product catalogs can potentially be generated using modern text-to-image models to populate image databases. This approach could improve the visual realism of synthetic environments without requiring large-scale manual data collection.

### 6.2. Fidelity of the Cloned Environment

VeriEnv is designed to synthesize functional training environments rather than pixel-perfect replicas of the original websites. In particular, our objective is to preserve the core interaction patterns and state transitions that are relevant for

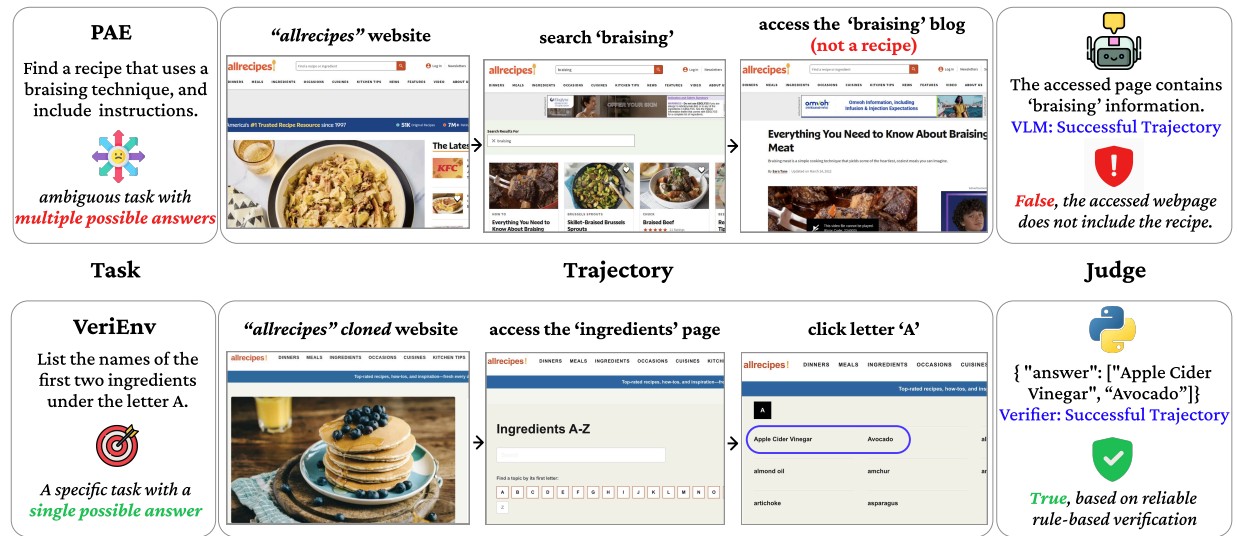

*Figure 6.* Comparison of task ambiguity and evaluation reliability in PAE (Zhou et al., 2025b) and VERIENV.

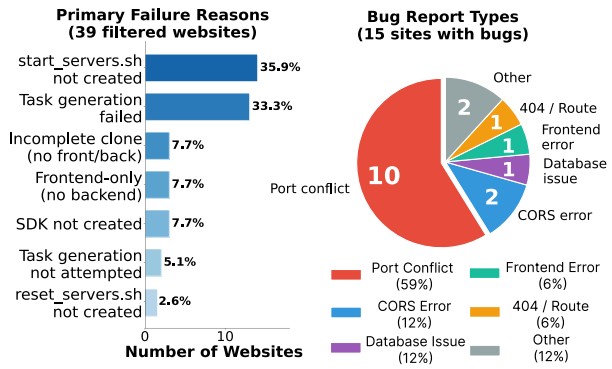

*Figure 7.* Primary failure reasons for websites excluded and bug types in error report.

web-agent training, while abstracting away website-specific details that are not essential for task execution. To assess whether the synthesized environments cover meaningful task functionality, we measure how many WebArena benchmark tasks can be supported by VeriEnv's Python SDK across three representative domains. Table 6 summarizes the coverage results. VeriEnv covers 88.6% of CMS tasks, 92.0% of Shopping tasks, and 95.4% of Map tasks, suggesting that the synthesized environments capture a large fraction of benchmark-relevant functionality.

The remaining uncovered tasks typically require features that are difficult to infer from limited visual observations alone. These include backend-specific reporting functions, such as generating coupon reports over a particular date range; account-level configuration changes, such as updating a user's address; and tasks requiring external geographic knowledge, such as answering which U.S. states border a given state.

*Table 6.* Coverage of WebArena benchmark tasks supported by VeriEnv's Python SDK across three representative domains.

| Metric | CMS | Shopping | Map |
| --- | --- | --- | --- |
| # WebArena Tasks | 184 | 187 | 109 |
| # Covered by VeriEnv | 163 | 172 | 104 |
| Coverage (%) | 88.6 | 92.0 | 95.4 |

### 6.3. Future Directions

An important direction for future work is to leverage these reconstructed environments for reinforcement learning. Because VERIENV provides executable and verifiable judges, the resulting reward signals are deterministic and reproducible, which substantially reduces the instability commonly observed in LLM-based or heuristic evaluation frameworks. We believe this setting enables a more principled study of self-evolving web agents, where agents continuously generate tasks, interact with environments, and improve through scalable training loops.

## 7. Conclusion

We presented VERIENV, which trains web agents in recreated websites by generating tasks with executable, verifiable validators. This design improves safety and reproducibility by avoiding interaction with real services and reducing reliance on subjective LLM judges. Experiments on WebArena and Mind2Web-Online show consistent gains over open-source baselines, and a site-specific setting demonstrates steady improvement through self-evolving training. Overall, our results support environment-centric scaling as a practical route to robust web agents.

## Impact Statement

This work aims to improve the safety, scalability, and reproducibility of web-agent learning by moving data generation and training into recreated websites. By enabling task creation with executable, verifiable validators, the approach can reduce dependence on subjective LLM-based judging and can facilitate more reliable benchmarking and ablation studies.

**Potential positive impacts.** Recreated websites can support rapid iteration for research on web agents without requiring repeated interaction with real services, which may lower the risk of unintended side effects such as spamming, policy violations, or accidental data modification. The use of deterministic validation can also improve experimental rigor and make agent-training pipelines easier to audit.

**Risks and negative impacts.** Techniques for recreating websites and training agents in high-fidelity web environments could be misused to develop agents that more effectively automate undesirable behaviors (e.g., large-scale scraping, account abuse, or manipulation of online services). Additionally, recreating websites could raise intellectual-property or terms-of-service concerns if used inappropriately, and recreated environments may inadvertently encode biased or unsafe content present in source sites.

**Mitigations.** Our framework emphasizes training on recreated environments rather than direct interaction with real services, and it relies on executable validators that can be designed to enforce safety constraints and limit harmful actions during training. To mitigate potential risks, all environments are executed in a sandboxed setting with external network access disabled. The SDK explicitly excludes payment flows, authentication mechanisms, and personally identifiable information, and agents interact solely through simulated browser actions. Internal state exposed via the SDK is used exclusively by the validator for post-hoc evaluation and is never accessible to the agent during execution.

We further encourage responsible use: cloning only websites for which permission is granted (or using internally created templates), limiting the fidelity of sensitive workflows, and releasing models and artifacts with appropriate safeguards (e.g., usage policies, rate limits, and evaluation focused on safety-critical behaviors). Finally, we recommend continued study of transfer from recreated environments to real deployment, including explicit safety evaluations before any real-world use.

## Acknowledgments

This research is supported in part by the NSF under grant numbers IIS-2052498, SMA-2418946, and NAIRR250217 Any opinions, findings, and conclusions or recommendations expressed in this material are those of the author(s) and do not necessarily reflect the views of the National Science Foundation.

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

# A. Implementation Details of VERIENV

## A.1. Agent Architecture and Training Hyperparameters

### A.1.1. CODING AGENT AND LLMS FOR IMPLEMENTING VERIENV

We experimented with several coding agent systems for implementing VERIENV, including Cursor CLI, Claude CLI, and OpenHands. In practice, we found that both Claude CLI and OpenHands frequently terminated the implementation process prematurely, even when the target website was not fully functional or when critical components such as the Python SDK were missing. These early exits made it difficult to reliably construct complete, production-ready synthetic environments.

We also evaluated alternative backbone language models for environment construction. In addition to GPT-5.2, which we use throughout our experiments, we tested open-source code-oriented LLMs such as Qwen3-Coder-30B-A3B-Instruct and GLM-4.7-Flash. However, we observed that the lack of strong multimodal capabilities significantly limited their effectiveness. In particular, these models struggled to diagnose and fix frontend issues (e.g., layout inconsistencies) and often failed to properly utilize available tools, such as executing shell commands or interacting with websites via Playwright MCP. As a result, they were less reliable for end-to-end website reconstruction in our setting.

### A.1.2. TRAINING DETAILS AND HYPERPARAMETERS

We train all web agents using LLaMA-Factory (Zheng et al., 2024b). For all experiments, we use a learning rate of $1 \times 10^{-5}$ and train for two epochs. We adopt a linear learning rate warmup over the first 10% of the total training steps.

Training is performed with a maximum sequence length of 8,000 tokens. We use DeepSpeed ZeRO-3 for memory-efficient distributed training, with a gradient accumulation step of 2. All experiments are conducted using two NVIDIA A40 GPUs.

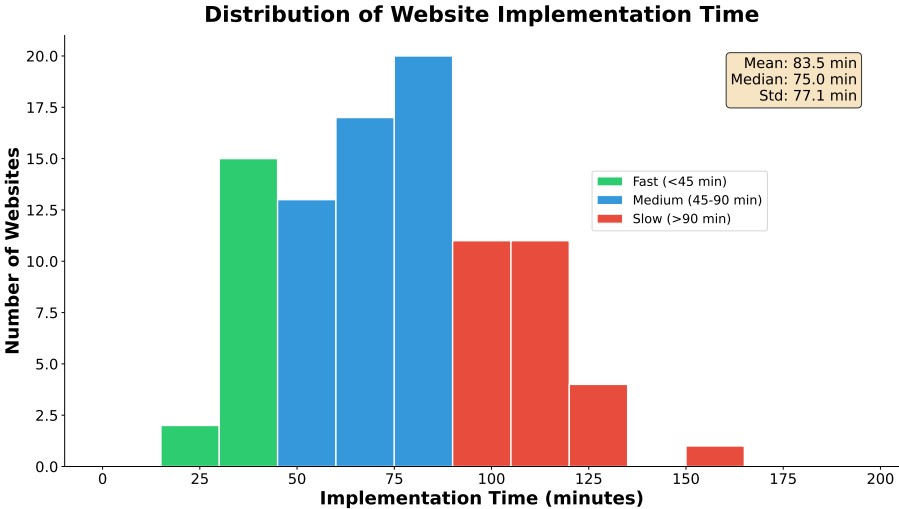

*Figure 8.* Distribution of website implementation time for constructing synthetic environments using a coding agent. Each bar shows the number of websites grouped by implementation duration, categorized as fast (¡45 minutes), medium (45–90 minutes), and slow (¿90 minutes). The distribution indicates that most websites can be reconstructed within a moderate time budget, with a long tail corresponding to more complex implementations.

## A.2. Synthetic Environment Construction Pipeline

We provide the prompt used for website reconstruction from snapshots in Figure 10.

We provide the prompt used by the coding agent for website implementation, bug reporting, and debugging in Figure 11.

## A.3. Task Generation and Validation Implementation

We provide the prompt used to generate tasks and validation judges in Figure 12.

**Example of Implementation and Debugging Process.** To illustrate the end-to-end implementation and debugging workflow enabled by VERIENV, we present a concrete example based on cloning a real-world retail website. Starting from a set of reference screenshots, the coding agent first constructs an initial executable version of the website, including frontend pages, backend APIs, database schemas, and a Python SDK that exposes internal functionalities. The initial implementation prioritizes functional completeness, ensuring that all major pages, navigation flows, and APIs are runnable via the provided server management scripts (e.g., `start_servers.sh`).

After the initial implementation, the coding agent performs iterative bug discovery using Playwright MCP. The agent systematically explores the synthesized website across different pages and viewports, comparing rendered content against the reference screenshots as in Figure 13. Discrepancies are documented as structured bug reports that include reproduction steps, expected versus actual behavior, and visual evidence captured as screenshots. The agent identifies issues such as missing homepage sections (Figure 14), incomplete long-form content on informational pages (Figure 15), and layout inconsistencies between desktop and mobile views (Figure 16).

Following bug reporting, the agent enters a debugging and patching phase. Reported issues are addressed by modifying frontend components, backend logic, and server scripts as needed. For instance, server lifecycle scripts are refined to ensure reliable resets between runs, and authentication-related bugs are fixed to correctly maintain session state across API calls. Visual and content-level refinements are applied to better align the synthesized website with the reference design. Each fix is verified through repeated Playwright-based testing, and the debugging outcomes are recorded in structured debug reports with post-fix screenshots as in Figure 17.

This example demonstrates how VERIENV supports a fully automated yet auditable environment construction pipeline, where implementation, bug discovery, and debugging are tightly coupled through executable scripts, visual inspection, and reproducible reports. Such iterative refinement is essential for producing high-fidelity synthetic environments that support verifiable task generation and reliable self-evolving agent learning.

### A.4. Cloned Synthetic Website Examples

We showcase examples of our cloned synthetic websites. Specifically, we randomly sample four sites (*e.g.*, , CarMax, CVS, Eventbrite, and Google Finance) and present representative screenshots from each site (Parts 1–3) illustrating key interface elements and interactions. We map each website's examples to the corresponding pages in Table 7.

*Table 7.* Cloned synthetic website screenshots. Each entry links to the corresponding figure for Parts 1–3.

| Website | Original URL | Part 1 | Part 2 | Part 3 |
|---|---|---|---|---|
| CarMax | https://www.carmax.com/ | Fig. 18 | Fig. 19 | Fig. 20 |
| CVS | https://www.cvs.com/ | Fig. 21 | Fig. 22 | Fig. 23 |
| Eventbrite | https://www.eventbrite.com/ | Fig. 24 | Fig. 25 | Fig. 26 |
| Google Finance | https://www.google.com/finance/ | Fig. 27 | Fig. 28 | Fig. 29 |

## B. Synthetic Website Evaluation Interface

This section documents the Label Studio[1] annotation interface used to assess (A) the quality of synthetic websites and (B) the validity of generated tasks and their associated validation programs ("judges"). Each annotation item provides the annotator with a website URL, a task instruction, and the judge code. Annotators interact with the website, record observed issues, and provide structured judgments. We

### B.1. Annotation Task

For each sample, annotators are given:

- **Website URL.** A link to the synthetic website instance.

- **Task instruction.** A natural-language description of the task the website is expected to support.

- **Task judge code.** A machine-checkable validator specification describing what constitutes task completion.

---

[1]https://labelstud.io/

Annotators complete two sections:

- **(A) Website Quality:** functional checks (feature-level checklist) and visual/appearance scoring (Likert scale).

- **(B) Task & Judge Validity:** binary judgments on whether the task is executable and whether the judge correctly evaluates completion.

**B.2. (A) Website Quality**

Website Quality separates *functional correctness* (whether features work) from *visual realism* (how the site looks), to reduce confounding between broken UI behavior and poor styling.

B.2.1. 1) CORE FUNCTIONAL CHECKS (CHECKLIST)

Annotators evaluate key interactive components of typical web services. For each functional area, annotators:

1. Test the feature on the website (e.g., attempt signup if available).

2. Select one status option.

3. Optionally add a brief description of distinct issues encountered.

Each functional check is a 3-way classification:

- **Works correctly:** the feature behaves as expected without functional errors.

- **Broken / Not working as expected:** the feature fails, produces errors, or behaves incorrectly.

- **Not applicable:** the website does not include the feature (e.g., no login form exists).

The checklist covers:

- **Signup / Registration** (if present).

- **Login** (if present; use provided test credentials if available).

- **Search functionality** (if a search bar or search UI exists).

- **Navigation & links** (menus, primary links, buttons leading to other pages).

- **Forms & submissions** (submit at least one form; verify success/error handling).

- **Filters / sorting / pagination** (if present in lists or search results).

B.2.2. 2) VISUAL / APPEARANCE (LIKERT SCALE)

Annotators assess the overall realism and visual quality of the website independent of whether features function. Visual issues include (but are not limited to) misaligned elements, broken layouts, missing images/icons, inconsistent styling, or clearly unfinished design.

**2.1 Overall Visual / Appearance Quality (1–5).**    Annotators select one rating based on the rubric below, counting *distinct* visual issues rather than repeated instances:

- **5 – Excellent:** 0–2 very minor visual issues; overall layout resembles a real-world website.

- **4 – Good:** 3–5 visual issues; minor inconsistencies but mostly realistic/professional.

- **3 – Fair:** 6–8 visual issues; noticeable problems but still understandable and usable.

- **2 – Poor:** 9–12 visual issues, or 1–2 severe visual failures (e.g., a key page is badly broken).

- **1 – Very Poor:** more than 12 visual issues, or multiple severe layout failures (e.g., overlapping sections, unreadable text).

## B.3. (B) Task and Judge Validation

This section evaluates whether the task is well-defined and executable on the site, and whether the judge correctly reflects true completion.

### B.3.1. INPUTS

Annotators are shown:

- **Task Instruction:** the natural-language task to attempt on the website (e.g., "search for a location and report rating and reviews").

- **Task Judge Code:** a validation specification (e.g., a set of required substrings such as a target rating and review count, with an evaluation type).

### B.3.2. BINARY JUDGMENTS

**1) Task Executability (Yes/No).** Annotators judge whether the task can be completed using the website as described:

- **Yes:** the task is doable with the website's available functionality and matches the instruction.

- **No:** the task is ambiguous, impossible, or depends on missing/non-functional components.

**2) Judge Correctness (Yes/No).** Annotators judge whether the validator accurately measures completion:

- **Yes:** the judge accepts correct completions and rejects incorrect ones, consistent with the instruction.

- **No:** the judge produces false positives/negatives or does not match the instruction semantics.

*Table 8.* Example tasks and evaluation criteria from VeriEnv environments.

| Website | Task Instruction | Judge Criteria | Diff. |
|---|---|---|---|
| adoptapet | Filter to cats and tell me how many results you get. | must_include: "2" | easy |
| airbnb | On the home page, what categories are available in the category row? Please list all of them. | must_include_all: "OMG!", "Lakefront", "Amazing pools", ... (11 items) | easy |
| allrecipes | Open the Ingredients A–Z directory and click the letter A. Tell me the first five ingredient names listed under A. | must_include: "Apple Cider Vinegar", "Avocado", "albacore tuna", "alfalfa", "almond oil" | easy |
| apartments | In Columbus, OH, sort listings by price (low to high). What is the first listing's name and minimum monthly price? | must_include: "The Charles at Bexley", "2938" | medium |
| bestbuy | Sign in, save two specific products, open Saved Items and tell me how many total saved items. | exact_match: "2" | easy |
| budget | Find the Red Ball Parking Garage location in New York, NY. Tell me the location code and hours. | must_include: "NYC1", "Mon-Sun 7:00 AM - 10:00 PM" | easy |
| careers.walmart | Filter jobs to Drivers & Transportation and pay type hourly. Tell me min and max pay. | must_include: "29", "36" | easy |
| carmax | Go to favorites page and tell me how many cars are in favorites list. | must_include: "1" | easy |
| coursera.org | Go to Resources, open 'Coursera Conference 2023', tell me the resource kind and summary. | must_include: "event", "Join leaders in higher education..." | easy |
| cvs | Sort by price low to high, find first in-stock product, tell me name and price. | must_include: "Dudley Group Vitamin C Gummies", "$9.99" | easy |

Table 8 – continued from previous page

| Website | Task Instruction | Judge Criteria | Diff. |
|---|---|---|---|
| discogs | Open first item under 'Trending Releases'. Tell me release title and artist name. | must_include: "Open-architected maximized Local Area Network", "Chavez Trio" | easy |
| drugs | Check interactions between 'Hydroxyzine' and 'Omeprazole'. Tell me severity and advice. | must_include: "minor" | hard |
| epicurious | Open 'Gluten-Free Cinnamon Crumb Cake' recipe, tell me cuisine and rating. | must_include: "Korean", "3.0" | easy |
| eventbrite | Check Tickets section. How many ticket types and cheapest price? | must_include: "2", "$15" | easy |
| exploretock | Open venue page for 'familiar formation'. Tell me address, city, state. | must_include: "220 Ash Street", "Chicago, IL" | easy |
| extraspace | Search for Tampa, FL. Tell me star rating and review count. | must_include: "4.7", "13" | easy |
| finance.google | Search for 'S&P 500', tell me ticker symbol and current value. | must_include: "SPX", fuzzy_match: "3746.49" | easy |
| finance.yahoo | Find Microsoft's ticker symbol. | must_include: "MSFT" | easy |
| foxsports | On Soccer league page, tell me first game's status. | must_include: "scheduled" | easy |
| gamestop | Find highest priced featured product. Tell me name and price. | must_include: "Nintendo Switch OLED Model - White" | easy |
| health.usnews | Search for 'keto'. Tell me type and title of first result. | must_include: "diet", "Keto Diet" | easy |
| ign | Go to deals page, tell me exact title of first deal. | exact_match: "Outside goal official defense..." | easy |
| instacart | In 'ALDI', find 'Basmati Rice - Family Size', tell me price. | must_include: "Basmati Rice - Family Size" | easy |
| jetblue | Search one-way JFK to SFO for 2/1. Tell me lowest and highest price. | must_include: "320.00", "432.00" | easy |
| linkedin | Log in, check Jobs alerts page, tell me how many alerts. | must_include: "3" | easy |
| target | Search 'softwaves'. Tell me total results and first product title. | must_include: "32", "Bluetooth Speaker" | easy |
| tesla | Filter Model 3 Used, sort by price. List first three prices. | must_include: "$31,805" | easy |
| thetrainline | From featured routes, find most expensive route (origin, destination, price). | must_include: "London", "Paris", "$43.57" | easy |
| uhaul | Create account, add items, remove one, tell me new subtotal. | must_include: "599" | easy |
| ups | Estimate shipping from 90012 to 92101. Tell me cost and delivery days. | must_include: "9.82", "4" | easy |

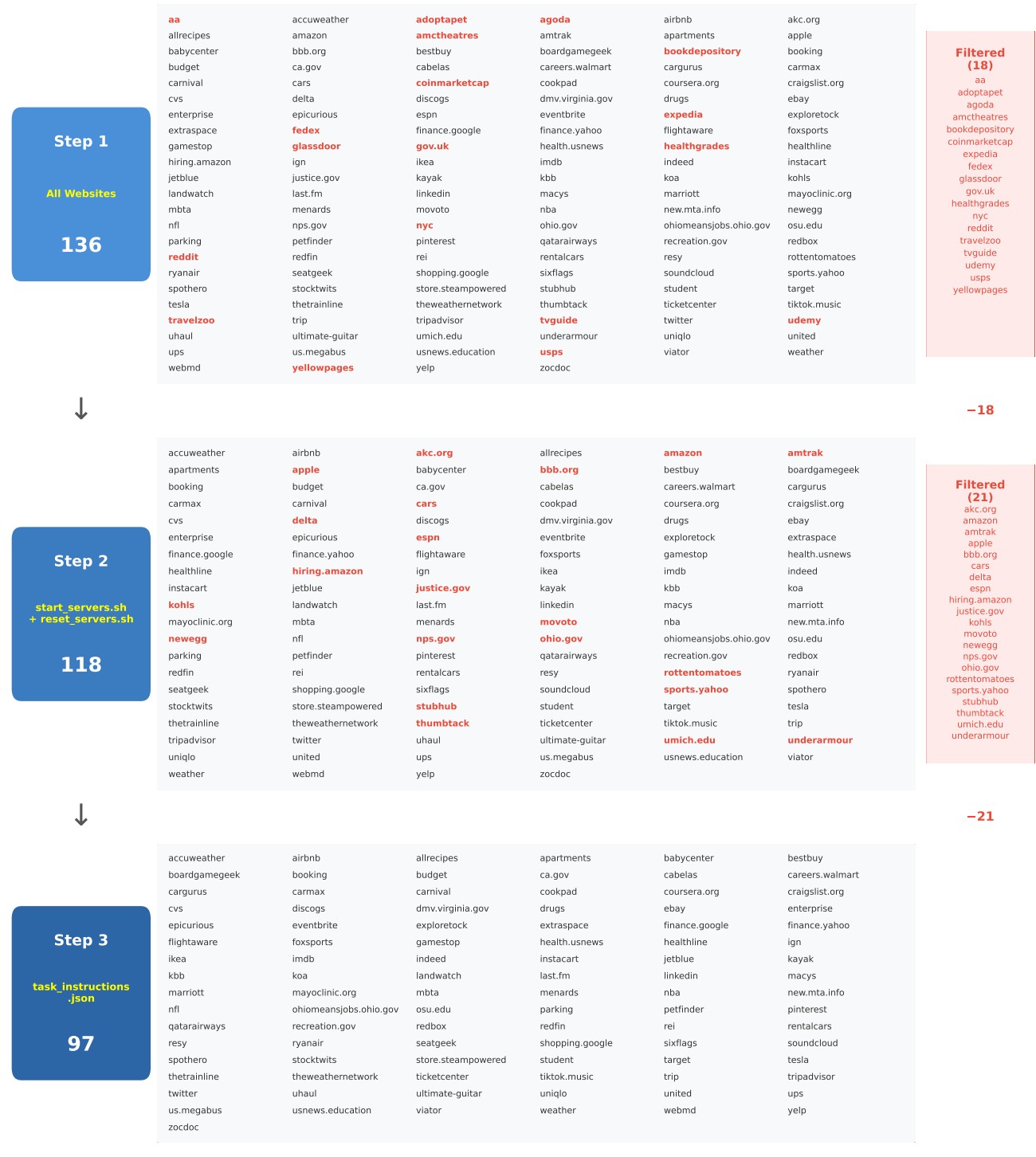

*Figure 9.* Website filtering flow for benchmark construction: 136 candidate websites are progressively filtered based on server script availability and task generation success, resulting in 97 validated benchmark websites.

**Prompts**

```
progress_tracking_and_documentation:  |
  Please use todo.md to track the progress of the implementation.
  Also, you have to document the implementation process in detail using markdown
format.
  Use Linear service to track the progress of the implementation.

implementation_prompt:  |
  You are an expert programmer that implements the given website in full production
quality.
  This is not a mockup website, so you need to implement all pages and features.

  Before implementing the website, inspect the ``landingpage.png'' and
``screenshot_x.png'' files
  and write a detailed and structured description of the website in
``website_description.md''.
  The implementation should be functionally and visually identical to the reference
website.
  For the database, populate realistic data instead of only a few dummy entries.

  You may use any tools and libraries you want, but the final website must be fully
deployable to production.
  For images, use assets from https://images.unsplash.com/.

  Also implement a Python SDK that can access the website's API, including
authentication,
  database access, and other related functions.

  You have to run the website locally and check if it is working correctly.
  If the website is not working correctly, fix the bugs and run it locally again.
  Repeat this process until the website is working correctly.
  You can use Playwright MCP to check the website and obtain screenshots that show
bugs or errors.

  After finishing the implementation, write a ``start_servers.sh'' script to start
the website locally.
  You need to run this script yourself.  If it is not working, fix the bugs and rerun
it.

  Lastly, implement a ``reset_servers.sh'' script to reset the servers.
  Many instructions will modify the database state, so ``reset_servers.sh'' should
restore the DB
  to its initial state.

  Please use only the pre-assigned ports in ``ports.json''.
  Do not use any other ports and do not kill any other processes using other assigned
ports.
```

*Figure 10.* Prompt used for website reconstruction by the coding agent.

**Additional Prompts**

```
general_prompt:  |
  We are building a production-ready clone of the website.
  You need to refer to the ''landingpage.png'' and ''screenshot_x.png'' files
  to implement a website that is fully functional and visually identical
  to the reference screenshots.
  In addition, implement a Python SDK that can be used to access the website's API,
  including the authentication process, database access, and related functions.

bug_report_prompt:  |
  Check whether the server is running correctly.
  Refer to the ''start_servers.sh'' file to start the server if needed.
  Ensure that ''reset_servers.sh'' correctly resets the server state to its initial
configuration.

  Analyze all functions and features of the website to identify bugs or errors.
  Use Playwright MCP to explore the website and capture screenshots that demonstrate
  the observed bugs or errors.  Save screenshots in the ''screenshot'' directory
  (create it if it does not exist) using the filename format
  bug_report_{timestamp}.png.

  You may report multiple bugs.  Visual similarity must also be considered by
comparing
  against ''landingpage.png'' and ''screenshot_x.png'' rather than images in the
screenshot directory.
  Do not leave any placeholder buttons or links.

  Document each bug in a markdown file named
  bug_report_{timestamp}.md.

debug_prompt:  |
  After reporting all identified bugs, debug the website and fix the issues.
  You are a professional programmer responsible for patching the website.
  First, fetch the issue reports from GitHub.
  Test the website functionality thoroughly to ensure correct behavior.

  After completing the patches, provide a detailed debugging report.
  You may again use Playwright MCP to verify fixes and capture screenshots.
  Save screenshots in the ''screenshot'' directory using the filename format
  debug_{timestamp}.png.
  Clearly explain the differences between the expected and actual results.

  Write the final debugging report to
  debug_{timestamp, yyyy-mm-dd_hh-mm-ss}.md.
```

*Figure 11.* Prompts used for website implementation, bug reporting, and debugging by the coding agent.

**Task and Judge Generation Prompt**

```
prompt:  |
 Please generate 50 diverse task instructions that can be conducted within this
website.
 Refer to the provided Python SDK when constructing the tasks.
 Each generated instruction must be fully validated using the Python SDK.

 The output file should be named task_instructions.json.
 Avoid synthetic or unrealistic instructions.

 The output JSON file should contain a list of instruction objects.
 Each object must include the following fields:

   - instruction:  A human-like task description (1--5 sentences).
   If authentication is required, the login process must be included.
   - python sdk tool call:  SDK calls used to verify the task.
   - tool call result:  The execution result of the SDK call.
   - is_valid:  Whether the instruction is valid.
   - difficulty:  One of easy, medium, or hard.

 Difficulty definitions:
   - Easy:  Simple browsing tasks with no authentication and minimal state changes.
   - Medium:  Multi-step tasks with navigation and limited stateful actions.
   - Hard:  Tasks requiring authentication and non-trivial state changes.

 judge_for_webagent format specification (IMPORTANT):
 The judge_for_webagent field must follow the exact JSON schema below.

 {
   "eval_type":  "rinfo" or "rprog",
   "parse":  "json" or null,
   "checks":  [ { "op", "expected", "path" (optional) } ]
 }

 Supported operations for checks:
   exact_match, must_include, fuzzy_match, must_include_all.

 Use eval_type = rinfo for answer-based evaluation and
 eval_type = rprog for programmatic verification (e.g., URL checks).
 Do not use deprecated string-based or natural-language judge formats.
```

*Figure 12.* Prompt used to generate verifiable task instructions and executable judges using the Python SDK.

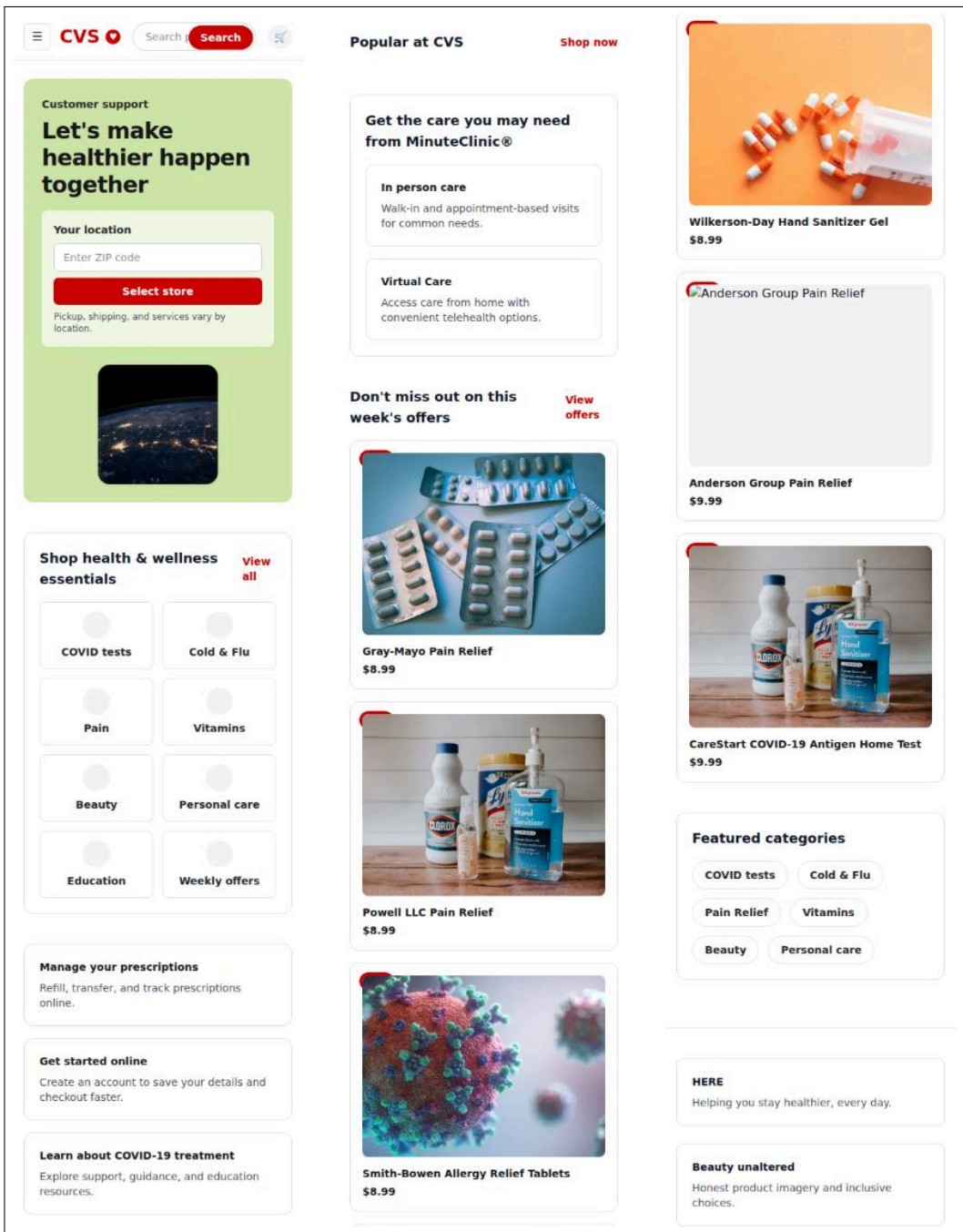

*Figure 13.* A browser snapshot attached to the bug report 'bug_report_2026-01-07_23-36-45'.

# Bug report (2026-01-07_23-34-16)

## Bug 1 — Homepage is not visually identical to reference

- **Reference**: `landingpage.png`
- **Actual screenshot**: `screenshot/bug_report_2026-01-07_23-34-16.png`

**Steps to reproduce**

1. Start the servers with `./start_servers.sh`
2. Open `http://localhost:12040/`

**Expected result**

The homepage should match `landingpage.png` (layout/sections/visual density), including the additional content blocks seen in the reference such as:

- Multiple "Popular at CVS"/promotional rails
- Additional "Get the care you may need…" content with image cards
- Expanded "This week's offers" area
- Large "Featured categories" + promotional banner area
- "Our commitments to you" section
- Full multi-column footer (with the larger set of links)

**Actual result**

The current homepage is a simplified subset:

- Several major sections present in `landingpage.png` are missing entirely.
- Page length/content density is far smaller than the reference.
- Footer/navigation content does not match the reference layout and link set.

*Figure 14.* Bug report example highlighting a missing homepage section discovered during automated traversal.

# Bug report (2026-01-07_23-35-05)

## Bug 1 — "At Home Covid Tests Education" content is incomplete vs reference

- **Reference**: `screenshot_1.png`
- **Actual screenshot**: `screenshot/bug_report_2026-01-07_23-35-05.png`

### Steps to reproduce

1. Start the servers with `./start_servers.sh`
2. Open `http://localhost:12040/covid/test-education`

### Expected result

The page should match `screenshot_1.png`, including the long-form educational content (many more sections/questions/answers, and the same overall page length and layout density).

### Actual result

The page renders only a shortened subset:

- The long FAQ + informational sections present in `screenshot_1.png` are missing.
- Overall page length is significantly shorter than the reference.

*Figure 15.* Bug report example showing incomplete long-form content on an informational page relative to the reference.

## Bug report (2026-01-07_23-36-45)

### Bug 1 — Mobile homepage layout/navigation does not match reference

- **Reference**: `screenshot_4.png`
- **Actual screenshot**: `screenshot/bug_report_2026-01-07_23-36-45.png`

**Steps to reproduce**

1. Start the servers with `./start_servers.sh`
2. Open `http://localhost:12040/` using a mobile viewport (e.g. 390×844)

**Expected result**

The mobile homepage should match `screenshot_4.png`, including:

- A mobile-specific header/navigation layout (hamburger + icon set and the left-side icon rail shown in the reference)
- Comparable content density and section ordering

**Actual result**

The current mobile layout is a simple stacked version of the desktop page:

- The reference's mobile navigation patterns (icon rail / mobile header structure) are missing.
- Several promotional/offer sections from the reference are missing, so the page is much shorter and visually different.

*Figure 16.* Bug report example illustrating a desktop–mobile layout mismatch detected across viewports.

## Debug report (2026-01-07_23-58-51)

### Fixed 1 — `reset_servers.sh` did not reliably stop the frontend (port 12040)

- **Expected**: Running `./reset_servers.sh` fully stops any running frontend/backend processes and frees ports **12040/12041**, so `./start_servers.sh` can always start cleanly.
- **Actual (before)**: Port **12040** could remain bound by an older `next-server` process. This caused `next start` to fail with `EADDRINUSE`, leaving the site partially broken (missing JS chunks / no hydration).
- **Fix**: Updated `reset_servers.sh` to kill listeners using `fuser` (with fallbacks) instead of relying on a single `ss` parsing path.

### Fixed 2 — `start_servers.sh` wrote an incorrect frontend PID and could start Next from the wrong working directory

- **Expected**: `./start_servers.sh` starts Next.js reliably and writes a PID that can be stopped later.
- **Actual (before)**:
- PID file could contain the wrapper shell PID, not the real Next process.
- After changing the startup command, Next could fail with "no production build" if started outside the `frontend/` directory.
- **Fix**: Start Next via a small shell wrapper that `cd` s into `frontend/` and `exec` s the `node .../next start` process, so the PID matches the running server process and `.next/` is found.

### Fixed 3 — Browser authentication did not persist, breaking "Orders"

- **Expected**: After signing in, visiting `/account/orders` should show the user's order history (or "No orders yet") without asking the user to sign in again.
- **Actual (before)**: Login could not persist in the browser (no token available for subsequent API calls), so `/account/orders` always rendered the "Please sign in…" state.
- **Fix**:

*Figure 17.* Structured debug report after patching, summarizing fixes and verification results.

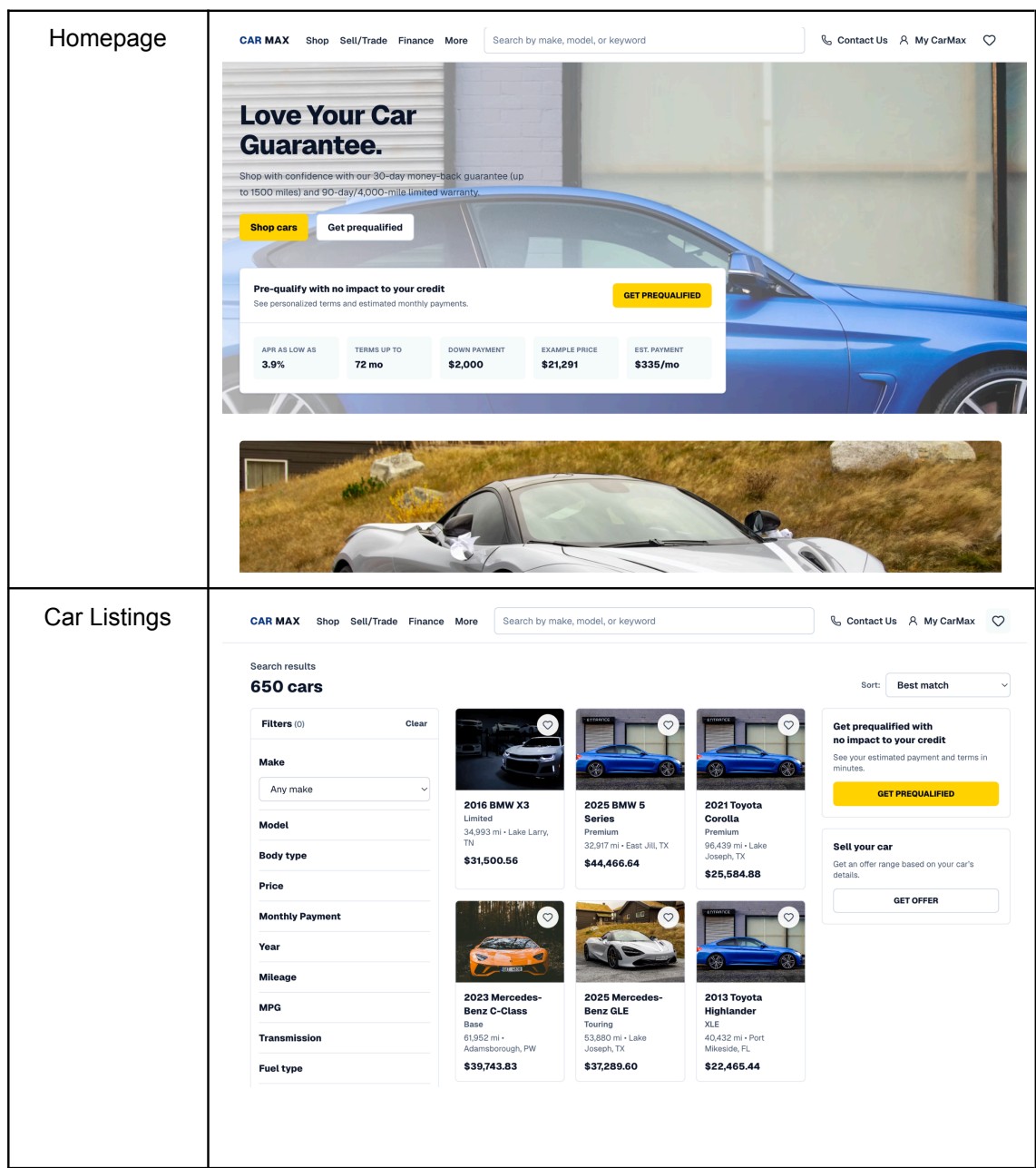

*Figure 18.* Screenshot from a cloned **CarMax** website (Part 1).

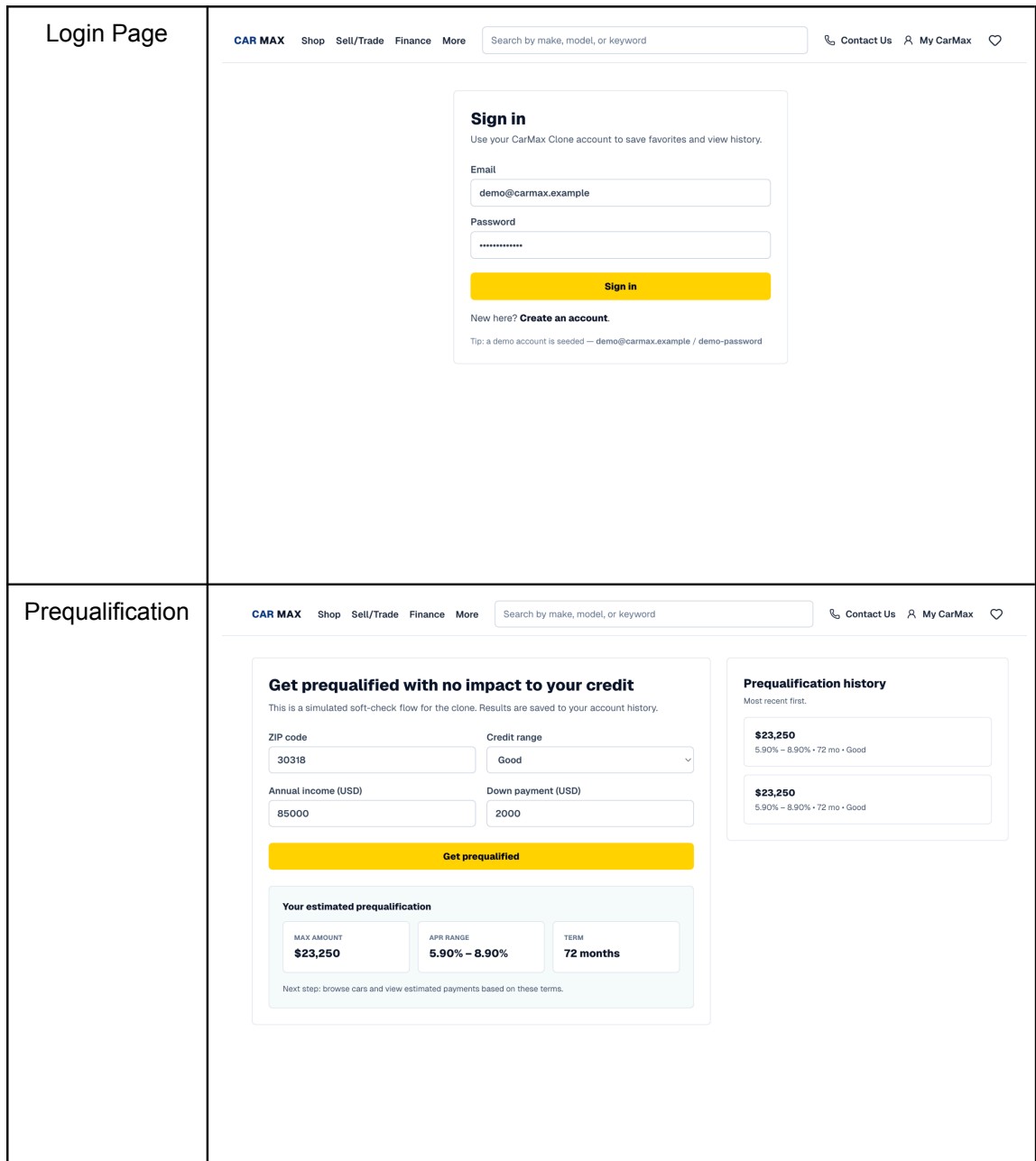

*Figure 19.* Screenshot from a cloned **CarMax** website (Part 2).

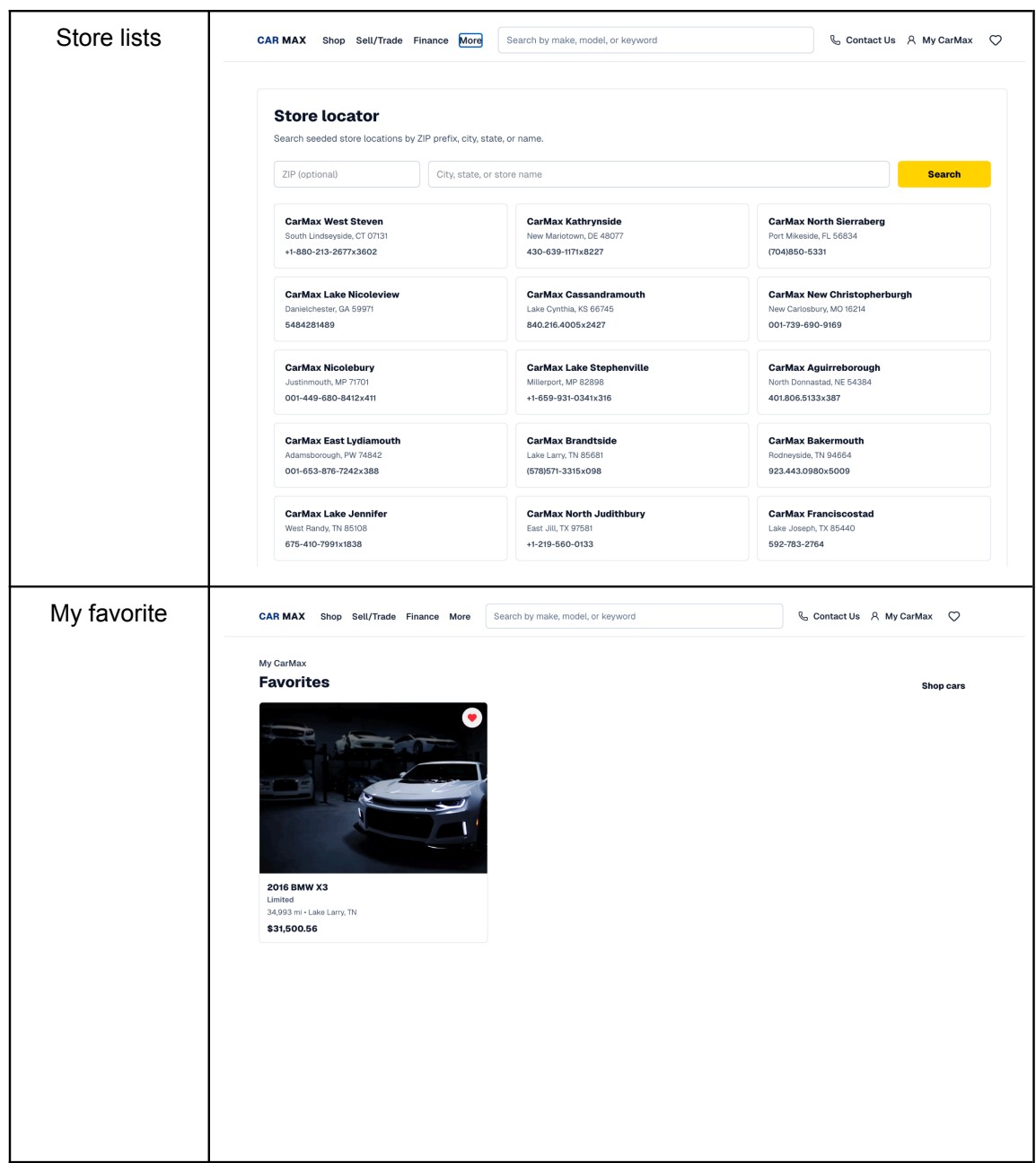

*Figure 20.* Screenshot from a cloned **CarMax** website (Part 3).

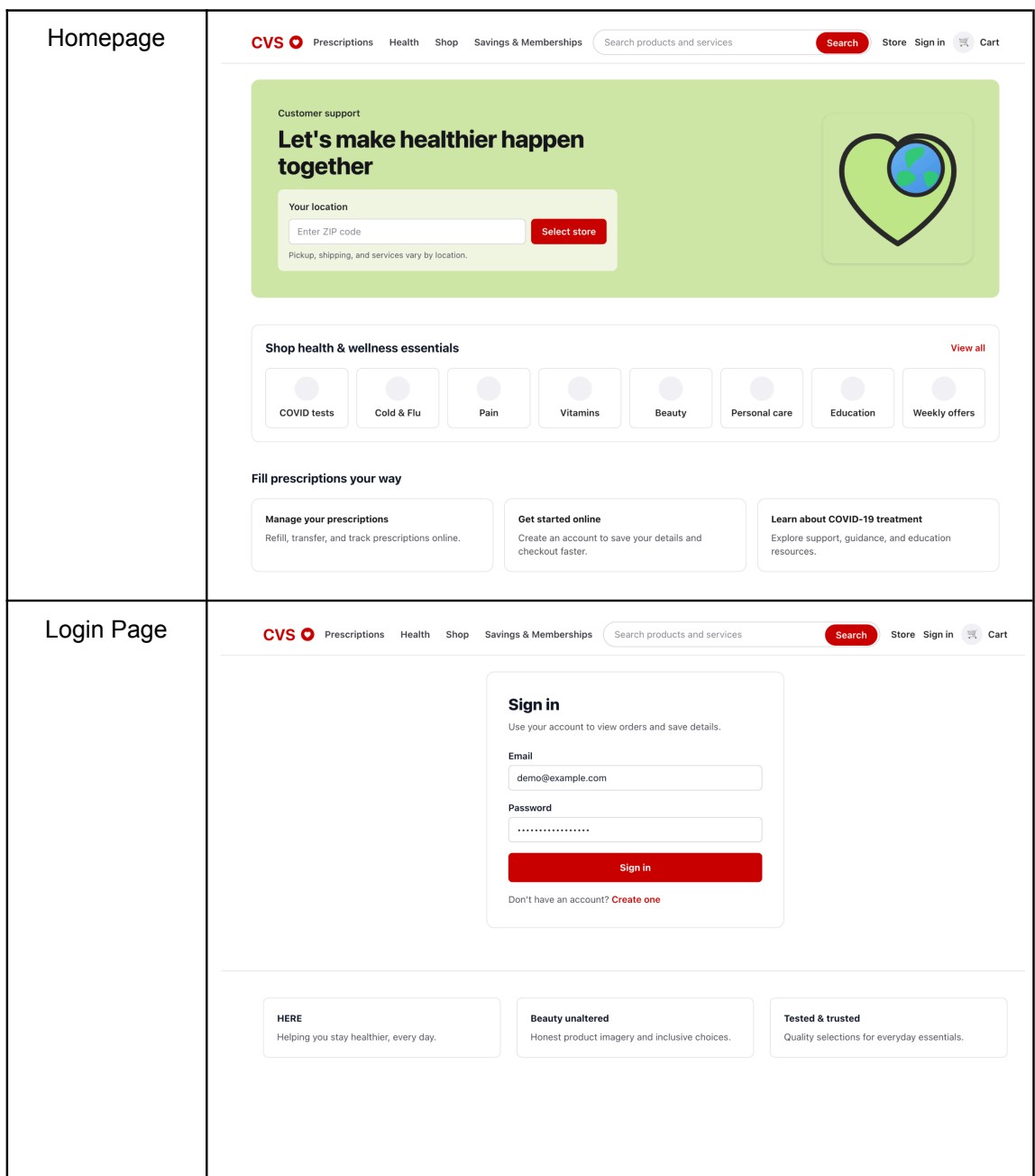

*Figure 21.* Screenshot from a cloned **CVS** website (Part 1).

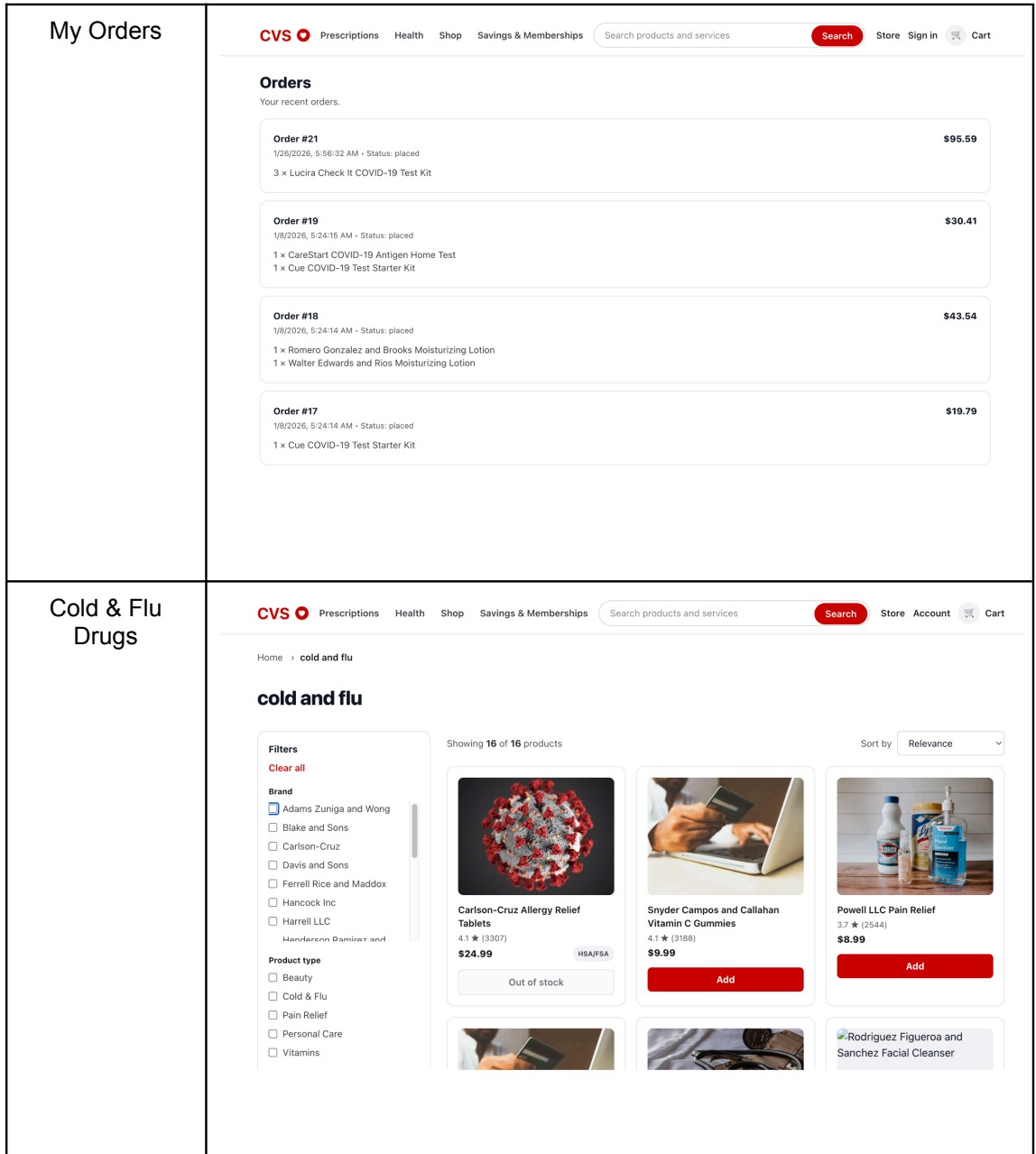

*Figure 22.* Screenshot from a cloned **CVS** website (Part 2).

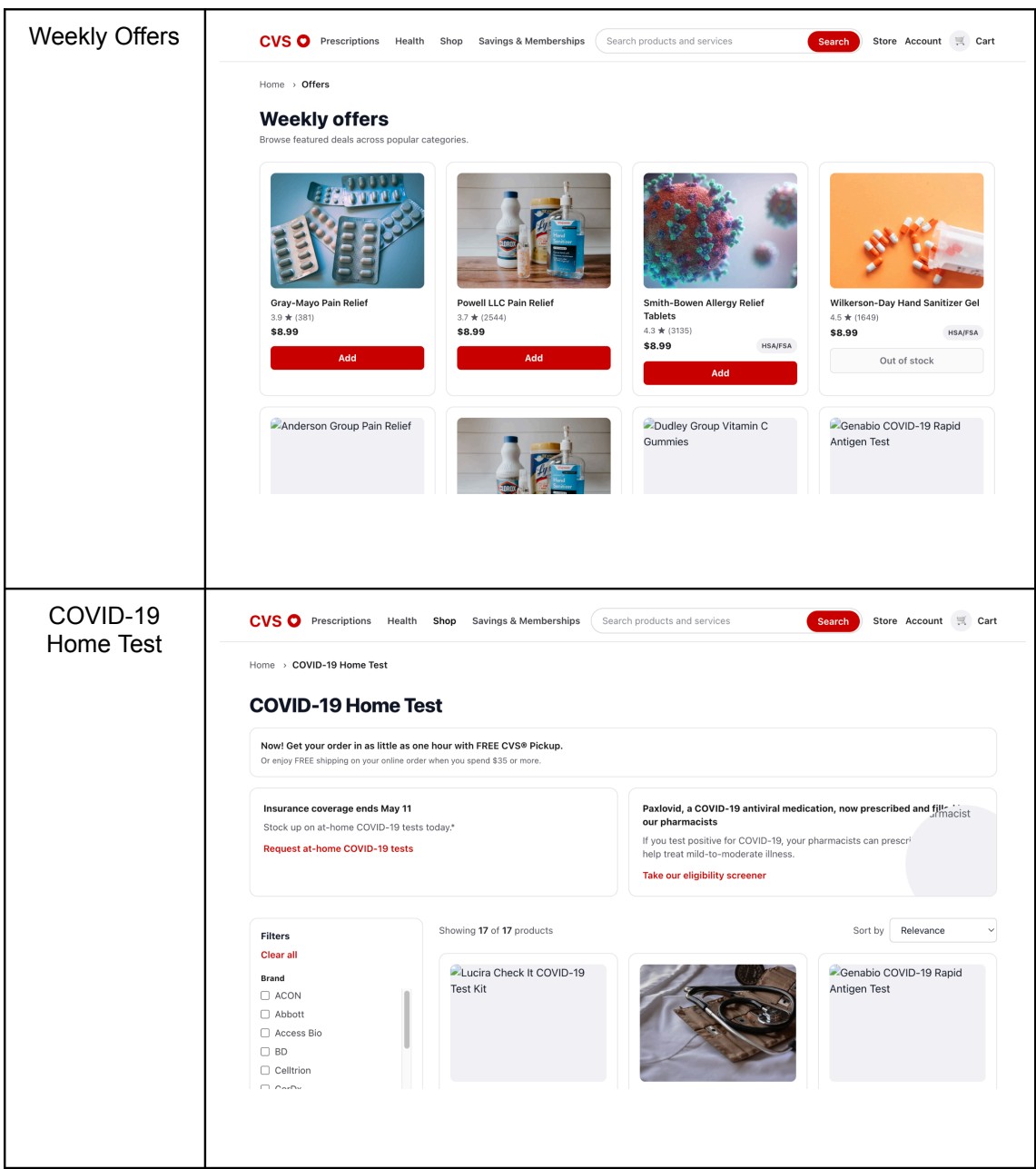

*Figure 23.* Screenshot from a cloned **CVS** website (Part 3).

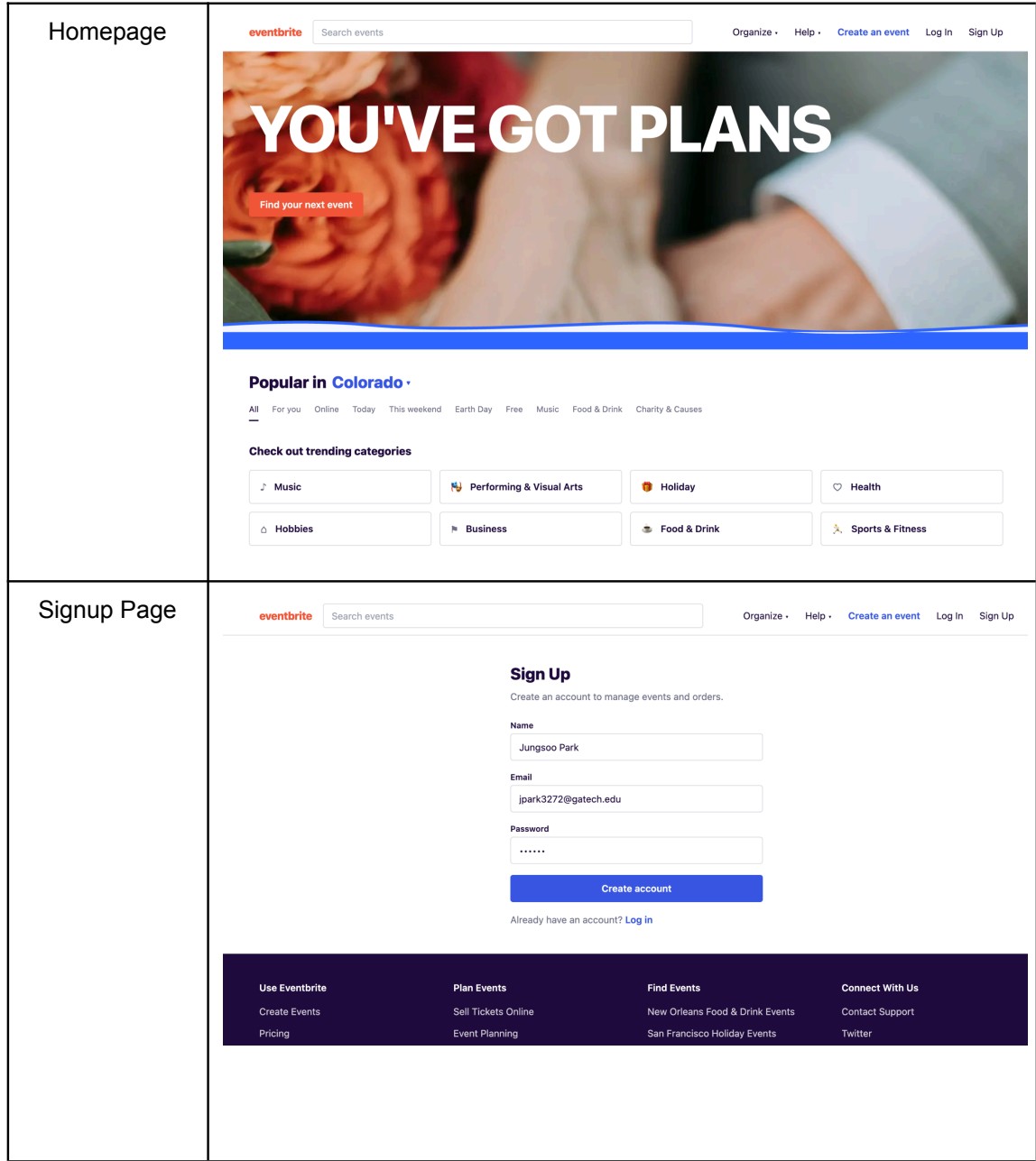

*Figure 24.* Screenshot from a cloned **eventbrite** website (Part 1).

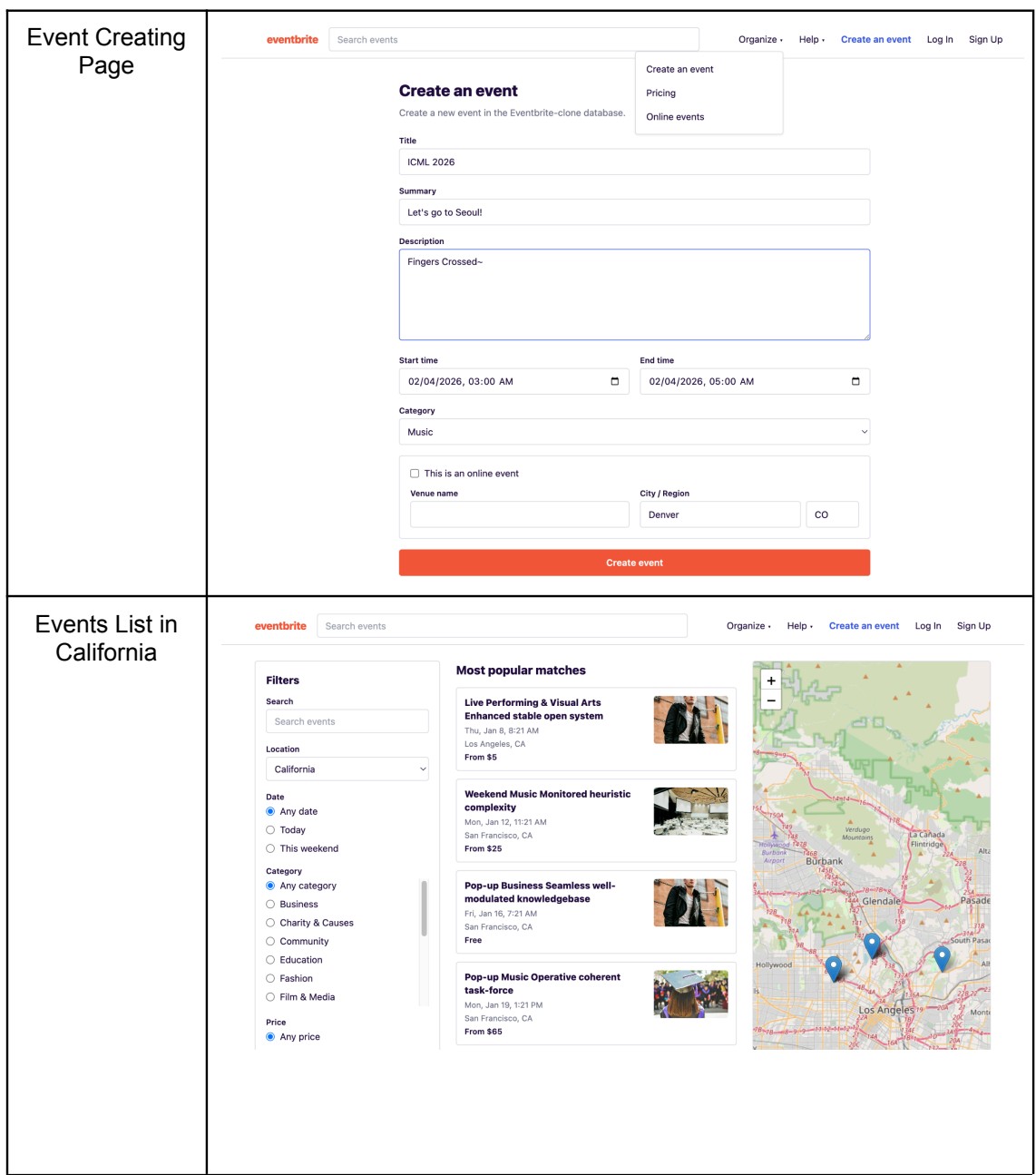

*Figure 25.* Screenshot from a cloned **eventbrite** website (Part 2).

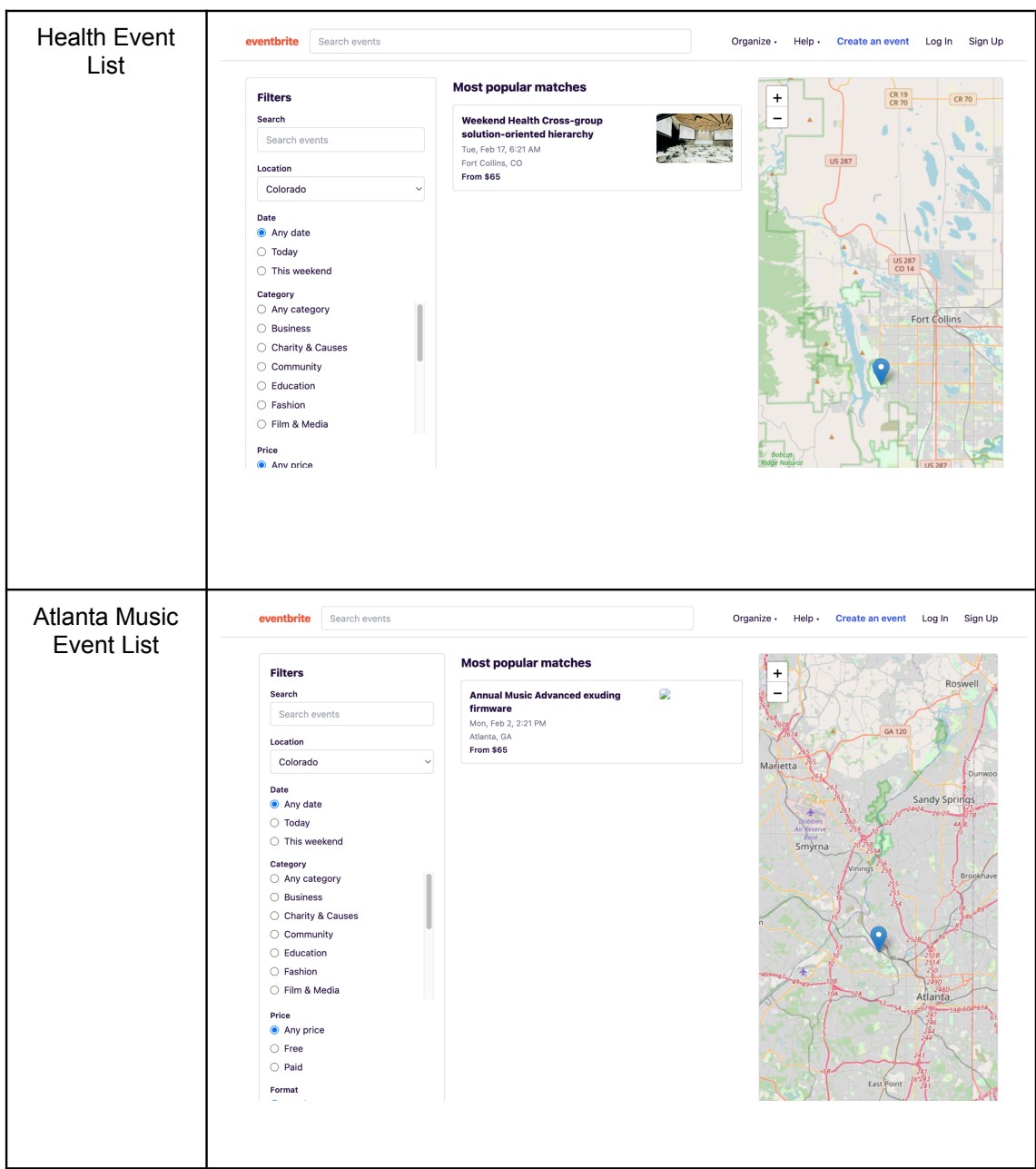

*Figure 26.* Screenshot from a cloned **eventbrite** website (Part 3).

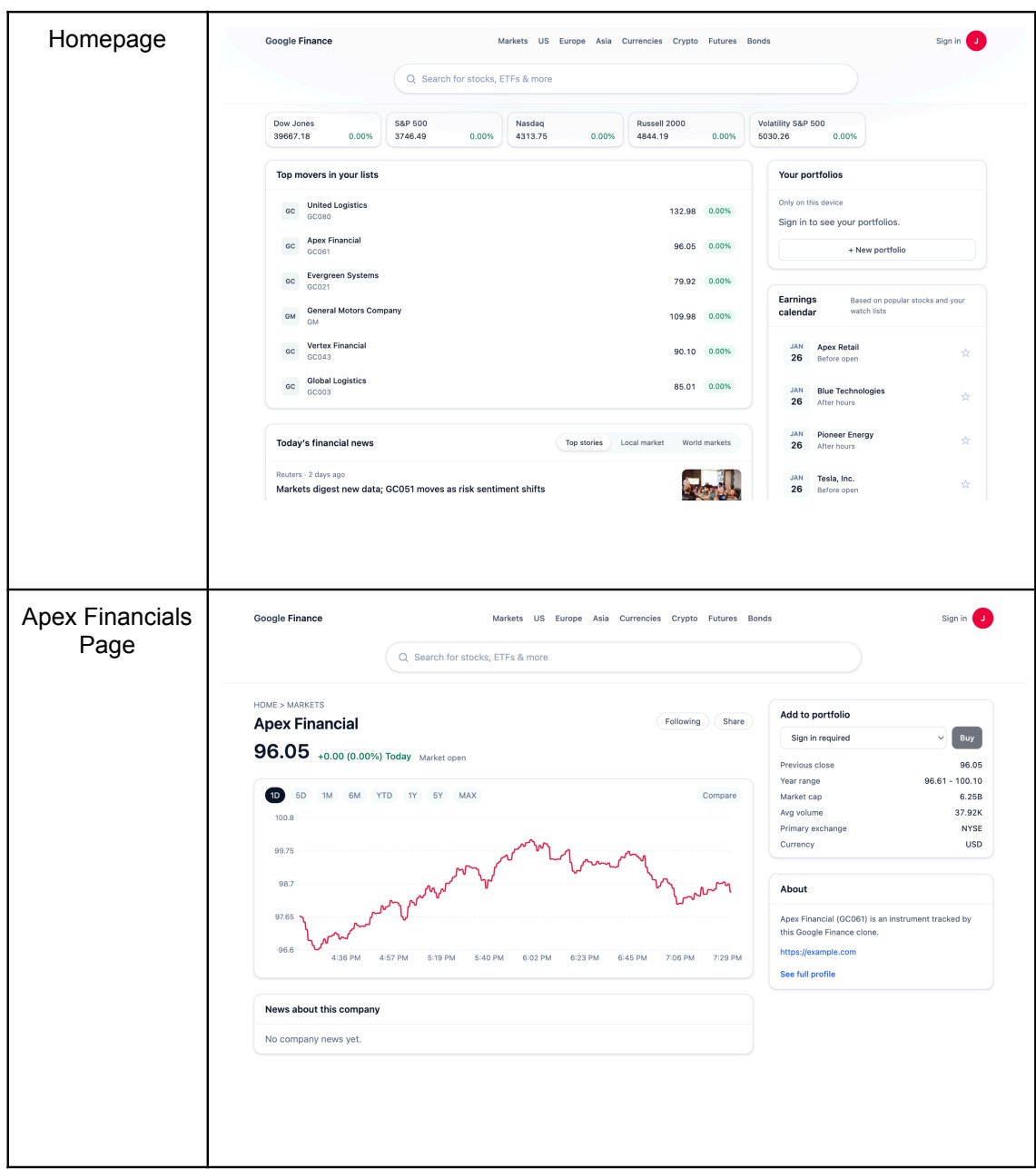

*Figure 27.* Screenshot from a cloned **Google Finance** website (Part 1).

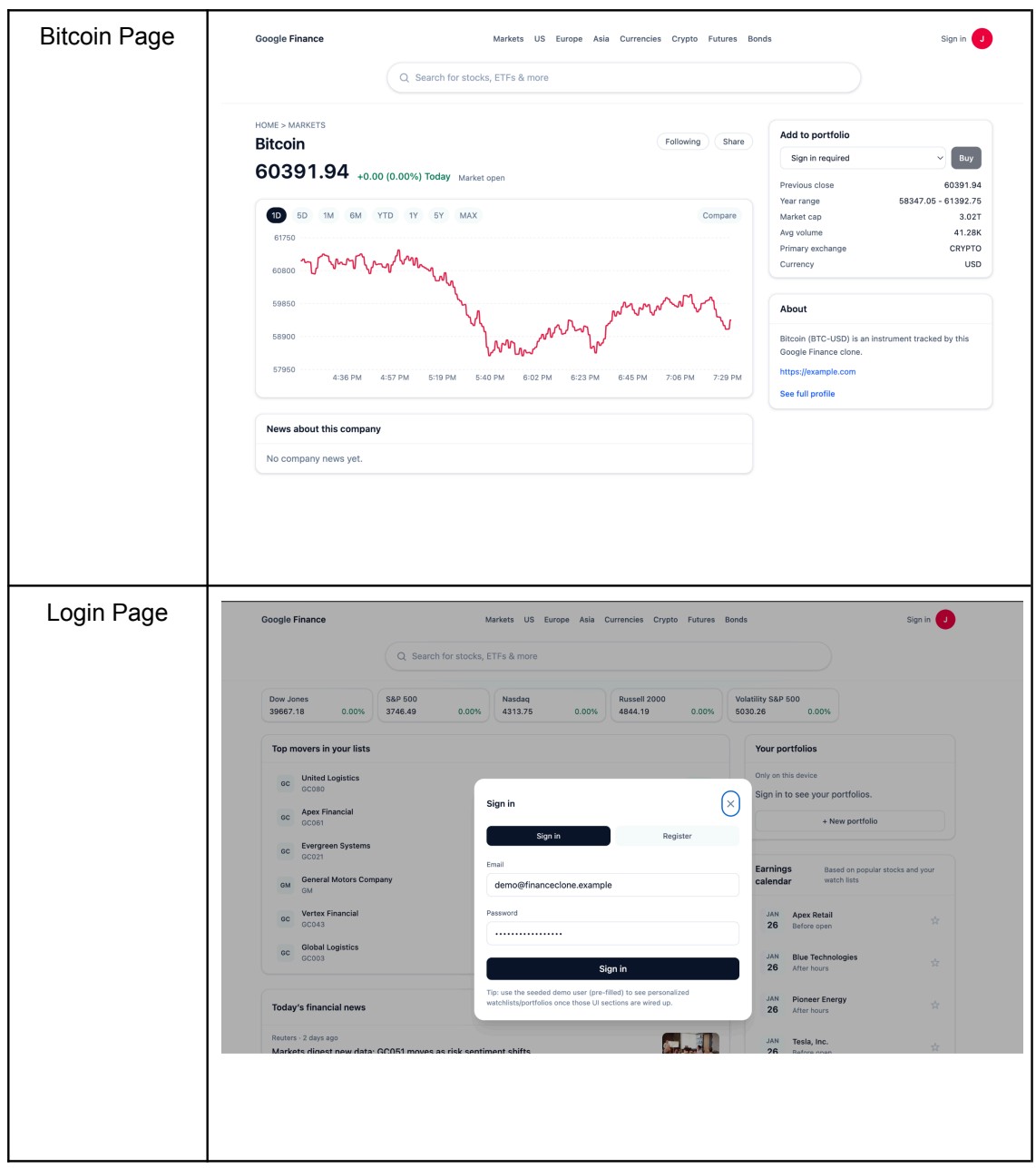

*Figure 28.* Screenshot from a cloned **Google Finance** website (Part 2).

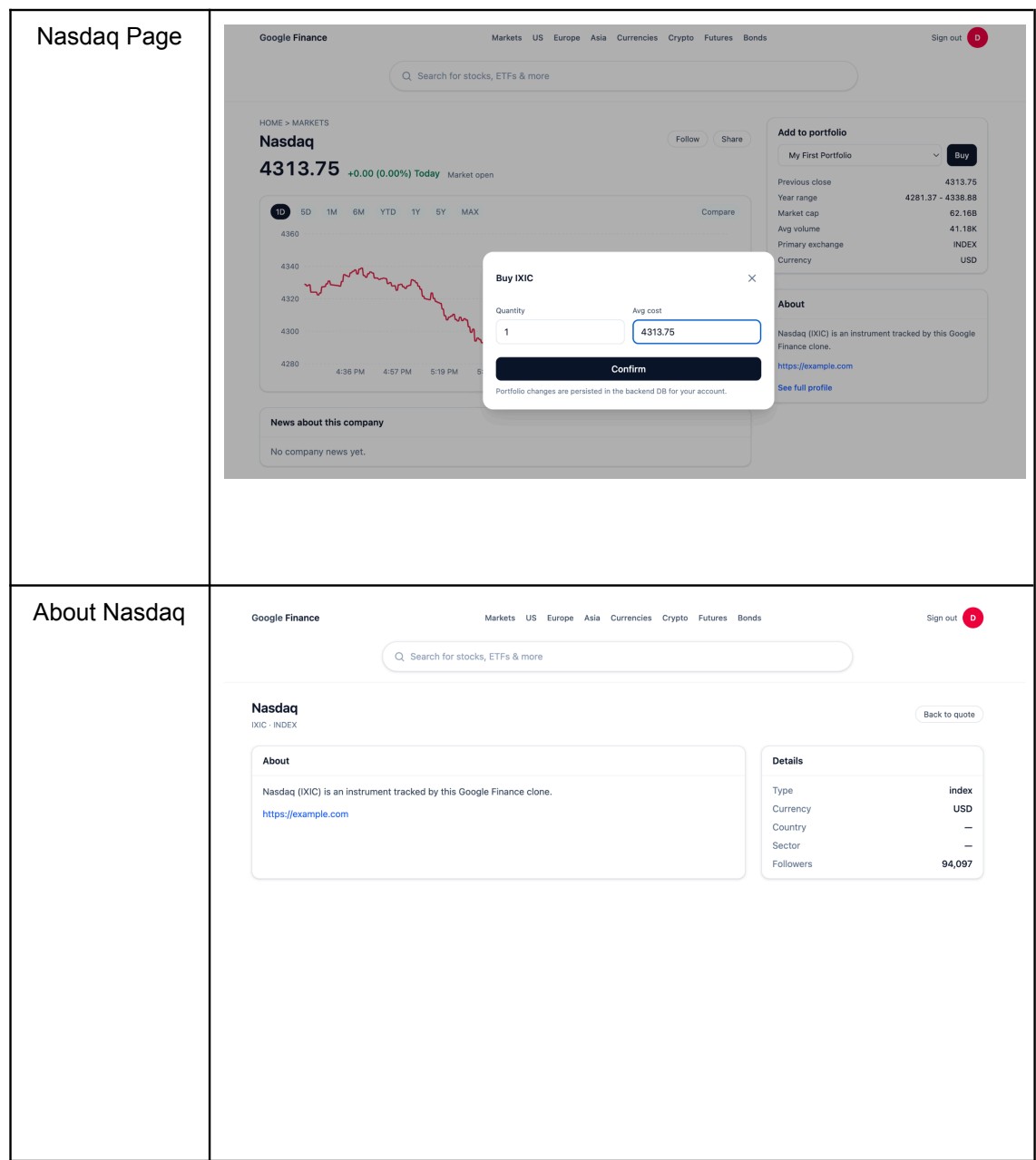

*Figure 29.* Screenshot from a cloned **Google Finance** website (Part 3).

