# OpenReview forum: "Safe and Scalable Web Agent Learning via Recreated Websites"
_ICML.cc/2026/Conference — ICML 2026 regular_

### Official Review · Reviewer_UGD2 · 2026-03-11

**Soundness:** 3
**Presentation:** 2
**Significance:** 2
**Originality:** 3
**Overall Recommendation:** 4
**Confidence:** 4

**Summary:**

This paper introduces VERIENV, a framework designed to overcome the safety, stability, and verifiability bottlenecks of training autonomous web agents on real-world websites. By employing a coding agent to automatically reconstruct target websites into fully executable synthetic environments the authors enable the self-generation of tasks accompanied by verifiable reward signals via a Python SDK. Through experiments on WebArena and Mind2Web-Online, the authors demonstrate that agents trained within these reconstructed environments achieve superior generalization to unseen sites and benefit significantly from increased environment scaling.

**Compliance With Llm Reviewing Policy:**

Affirmed.

**Final Justification:**

Most concerns are resolved

**Key Questions For Authors:**

No

**Strengths And Weaknesses:**

Summary of Strengths:
1. VERIENV introduces a novel framework that automatically clones real-world websites into fully functional synthetic environments, including backend logic and database states, effectively decoupling agent learning from unsafe, restricted live-web interactions.
2.  The experiments demonstrate significant performance improvements on WebArena and Mind2Web-Online.

Summary of Weaknesses:
1. While the authors claim full cloning, the environments remain local approximations of real websites. There is no rigorous quantitative analysis on the divergence of agent behavior between the cloned environment and the live production environment
2. The paper asserts the importance of verifiable rewards but lacks a direct, controlled ablation comparing their proposed "Python SDK-based judge" against a state-of-the-art "LLM-as-a-Judge" on the exact same synthetic tasks.
3. In the comparison against PAE, the authors do not demonstrate that both frameworks utilize an identical number or diversity of training tasks, leaving it unclear whether performance gains are due to the method or simply data scale.
4. The framework lacks a clear explanation of how the coding agent interprets website "intent" from screenshots, and the paper fails to explicitly clarify whether the cloning agent is conditioned on a single static image or a comprehensive sequence of interaction traces.

---

> ### Author Rebuttal · Authors · 2026-03-31
>
> We thank the reviewer for the detailed and constructive review.
>
> **W1. Quantitative analysis on the divergence between cloned and real environments**
>
> Thank you for pointing out the lack of rigorous quantitative analysis. To directly measure how much the cloned environments diverge from the original websites, we conducted a task-level coverage analysis: for each WebArena domain, we checked how many benchmark tasks can be executed and verified through VeriEnv's Python SDK.
>
> | Domain | \# Tasks in WebArena Benchmark | \# Covered by VeriEnv’s Python SDK | Coverage (%) |
> | :---- | :---- | :---- | :---- |
> | CMS | 184 | 163 | 88.6% |
> | Shopping | 187 | 172 | 92.0% |
> | Map | 109 | 104 | 95.4% |
>
> The coverage ranges from 88.6% to 95.4%, demonstrating that VeriEnv captures the vast majority of tasks in these domains. The uncovered tasks tend to require features that are difficult to infer from screenshots alone, such as generating specific backend reports (e.g., "Create a coupons report from 05/01/2021 to 05/15/2023"), modifying account-level settings (e.g., "update my address on OneStopMarket"), or answering geographic knowledge questions (e.g., "Which US states border Massachusetts?").
>
>
>
> **W2. Comparing the LLM judge and Python SDK-based judge**
> We agree that a direct comparison is important. We conducted a human evaluation where evaluators on Prolific annotated web agent trajectories as ground truth, and compared our SDK judge against an LLM judge (WebJudge-7B):
>
> | Judge Type | Accuracy (%) | TP (%) |  FN (%) | TN (%) | FP (%) |
> | :---- | :---- | :---- | :---- | :---- | :---- |
> | LLM Judge (WebJudge-7B) | 62.3 | 31.9 | 33.3 | 30.4 | 4.3 |
> | Python SDK Judge (Ours) | 94.2 | 60.9 | 4.3 | 33.3 | 1.4 |
>
>
>
> The SDK judge achieves 94.2% accuracy with an FP of only 1.4%, meaning it rarely includes incorrect trajectories as training data. The LLM judge achieves only 62.3% accuracy, with a 3× higher FP (4.3% vs. 1.4%) and a much higher FN (33.3% vs 4.3%) — it both mislabels more failures as successes and discards more genuinely successful trajectories. The SDK judge is also deterministic (100% inter-run consistency) and orders of magnitude cheaper to run.
>
> Critically, this difference in judge quality directly impacts downstream training. We verify this through an ablation where we swap only the judge while keeping everything else fixed:
>
> | Configuration (Base model: Qwen3-4B) | WebArena | Mind2Web-Online |
> | :---- | :---- | ----- |
> | (A) Base model (no training) | 7.88 | 13.18 |
> | (B) SFT on all VeriEnv trajectories (no filtering) | 2.42  | 9.55 |
> | (C) RFT with LLM judge on VeriEnv | 9.09 | 15.91 |
> | (D) RFT with SDK judge on VeriEnv | 13.94  | 20.45 |
>
> Training without any filtering (B) actually degrades performance below the base model (7.88→2.42), confirming that indiscriminate trajectory inclusion is harmful. The LLM judge (C) recovers to 9.09/15.91 but remains well below the SDK judge (D) at 13.94/20.45 — a gap of \+4.85 on WebArena and \+4.54 on Mind2Web-Online. This demonstrates that the SDK judge's accuracy advantage translates directly into better agent performance.
>
> **W3. Clarification on the experimental setup**
>
> To ensure a fair comparison, we utilized an identical number of 500 tasks for each website during all experiments comparing VeriEnv against PAE. Both settings were trained for the same number of iterations with the same base model. We appreciate you raising this point, and we will ensure this detail is clearly stated in the camera-ready version.
>
> **W4. Details of how VeriEnv understands the implementation task**
>
> To clarify, the cloning agent is conditioned on a sequence of 5 screenshots (not a single static image), taken from the first five steps of a Mind2Web trajectory. These capture key interaction states (e.g., homepage → search results → item detail → form submission → confirmation), giving the agent a sense of the website's navigation flow and behavior.
>
> The coding agent interprets website intent through a two-stage reasoning process before writing any code:
>
> 1. **Understanding website intent**: The agent first reasons about what this website is for and what users can do on it — identifying the domain (e.g., e-commerce), core workflows (e.g., browse → filter → add to cart → checkout), and supporting UI components. It writes this into a structured `website_description.md` that serves as the implementation specification.
> 2. **Architecture planning → Implementation → Self-debugging**: Based on the description, the agent designs the technical architecture (database schema, API endpoints, frontend routes, Python SDK), generates full-stack code, and iteratively refines it by deploying the site and comparing rendered output against the original screenshots via Playwright (Section 3.2).
> In short, the agent reasons about the website's purpose and user-facing behavior from the screenshot sequence, then builds a functionally equivalent environment, rather than reverse-engineering the original website.

---

> > ### Author Rebuttal · Reviewer_UGD2 · 2026-04-02
> >
> > Thank you for the response.
> > At least half of my concerns are addressed or partially addressed.
> > I tend to maintain my initial score, as it is already positive, and I believe my judgment is fair.
> > To recognize the authors' effort, I will raise my confidence score from 3 to 4.

---

> > > ### Author Response · Authors · 2026-04-03
> > >
> > > Thank you for your thoughtful feedback and for considering our rebuttal. We truly appreciate your positive evaluation and are grateful that you recognized our efforts by increasing your confidence score.

---

### Official Review · Reviewer_t6B2 · 2026-03-12

**Soundness:** 3
**Presentation:** 3
**Significance:** 3
**Originality:** 2
**Overall Recommendation:** 5
**Confidence:** 3

**Summary:**

This paper presents VeriEnv, a framework used to create synthetic reproductions of real websites using screenshots. The framework also enables the creation of synthetic tasks and verifiers to train web agents to interact with web platforms without accessing the real system. This enables safe and reproducible training of web agents as the environments can be used offline, and the verifiers can offer deterministic rewards instead of relying on an LLM-as-a-judge setup.

**Compliance With Llm Reviewing Policy:**

Affirmed.

**Final Justification:**

VeriEnv is well-motivated and technically solid. The rebuttal resolved my main concerns: coverage analysis quantifies environment fidelity, the real-time SDK judge achieves 94.2% accuracy (explaining the 76% figure in Table 2), and the ablation confirms gains come from reward quality rather than data volume. Originality is somewhat incremental but the practical contribution is clear. Raising my score to 5.

**Key Questions For Authors:**

1. How much of the observed performance gain comes specifically from verifiable reward signals versus simply generating additional training trajectories?

**Limitations:**

yes

**Strengths And Weaknesses:**

### Strengths

- The paper is well motivated and convincingly highlights key challenges in training web-based agents, including unsafe and unreliable exploration, brittle validation via LLM-as-a-judge, and complexity of resetting real environments. Their proposed framework directly addresses these issues.

- The paper provides useful engineering insights, including environment construction statistics, human evaluation of cloned sites, and error analyses of environment generation failures.


### Weaknesses

- **Unclear fidelity of cloned environments.** The paper mentions that "web environments exhibit a much smaller discrepancy between synthetic and real executions when the underlying functionality and state transitions are faithfully reproduced" (L367-370). However, given that the coding agents can only reproduce the environment via screenshots, it does not have full observability of the real website. It is unclear how faithful these transitions are given that we don't take

- **Unreliable validators.** The authors mention that their approach provides better verifiability compared to LLM-as-a-judge setups, but the validators that are automatically are not that reliable either (~76% judge correctness, Table 2). They mention that this is due to some randomness in the way the database of the websites is populated. Both of these indicate that the proposed method might be less robust than thought.

- **Unclear where the benefits come from.** The proposed framework introduces two changes simultaneously: (1) replacing real websites with recreated synthetic environments, and (2) replacing LLM-based evaluation with executable, verifiable validators. However, the evaluation does not isolate the contribution of these components. It remains unclear whether the performance gains arise from the verifiable reward signals, the use of synthetic environments, or simply from the additional training data generated in the process. For example, one could imagine mining trajectories directly from real websites and attaching verifiable validators without recreating the environment. An ablation disentangling these factors would significantly strengthen the claims of the paper.

---

> ### Author Rebuttal · Authors · 2026-03-31
>
> We thank the reviewer for the thoughtful and constructive review.
>
> **W1. Fidelity of the cloned environment**
>
> We agree that fully replicating original websites from a few screenshots alone is very challenging. We would like to clarify that VeriEnv's goal is **not** to create a 100% identical copy of the original website, but rather to build a functional environment that captures the core interaction patterns and state transitions relevant to agent training. If our draft conveyed otherwise, we will revise the language accordingly.
>
> To quantify the coverage of the synthesized environments, we analyzed how many WebArena benchmark tasks can be covered by VeriEnv's Python SDK across three representative domains:
>
> | Domain | \# Tasks in WebArena Benchmark | \# Covered by VeriEnv’s Python SDK | Coverage (%) |
> | :---- | :---- | :---- | :---- |
> | CMS | 184 | 163 | 88.6 |
> | Shopping | 187 | 172 | 92.0% |
> | Map | 109 | 104 | 95.4 |
>
> The coverage ranges from 88.6% to 95.4%. The uncovered tasks tend to require features difficult to infer from screenshots, such as generating specific backend reports (e.g., "Create a coupons report from 05/01/2021 to 05/15/2023"), modifying account-level settings (e.g., "update my address"), or answering geographic knowledge questions (e.g., "Which US states border Massachusetts?").
>
>
>
> **W2. Human evaluation of judge correctness**
>
> We appreciate the concern about the 76% judge correctness. We note that this number reflects a specific evaluation setup that does not fully represent the judge used during training. In the original human evaluation (Table 2), we served websites on a separate external server for annotators, which caused a misalignment between the database state accessed by the SDK judge and the one presented to annotators. We refer to this as the "static" SDK judge.
>
> To provide a more rigorous analysis, we conducted a new evaluation where human evaluators on Prolific directly annotated web agent trajectories as ground truth. We then compared the configurations:
>
> | Judge Type | Accuracy (%) | TP (%) |  FN (%) | TN (%) | FP (%) |
> | :---- | :---- | :---- | :---- | :---- | :---- |
> | LLM Judge (WebJudge-7B) | 62.3 | 31.9 | 33.3 | 30.4 | 4.3 |
> | Python SDK Judge (Ours) | 94.2 | 60.9 | 4.3 | 33.3 | 1.4 |
>
> The real-time SDK judge — which is what we actually use during training — achieves 94.2% accuracy with an FP of only 1.4%, confirming that it rarely includes incorrect trajectories as training data. The 76% figure from Table 2 is an artifact of the static evaluation setup (misaligned database state on the external server), not a reflection of training-time judge quality.
>
> In contrast, the LLM judge achieves only 62.3% accuracy, with a 3× higher FP (4.3% vs. 1.4%) and a much higher FN (33.3% vs. 4.3%). The LLM judge evaluates trajectories based on surface-level visual appearance, making it unreliable for distinguishing genuinely successful completions from plausible-looking failures. For rejection fine-tuning, this directly degrades training quality — both by including bad trajectories and discarding good ones.
>
> **W3 & Q1. Ablation: where do the benefits come from?**
>
> Both the weakness and the key question ask the same core question: how much of the gain comes from verifiable reward signals versus simply having more training data? We agree that disentangling these contributions is important, and we conduct the following ablation study. Note that all configurations use rejection fine-tuning (i.e., filtering successful trajectories for SFT), which is the training method used throughout our paper.
>
> | Configuration (Base model: Qwen3-4B) | WebArena | Mind2Web-Online |
> | :---- | :---- | ----- |
> | (A) Base model (no training) | 7.88 | 13.18 |
> | (B) SFT on all VeriEnv trajectories (no filtering) | 2.42  | 9.55 |
> | (C) Rejection FT with LLM judge on VeriEnv environments | 9.09 | 15.91 |
> | (D) Rejection FT with SDK judge on VeriEnv | 13.94  | 20.45 |
>
> Key findings:
>
> * **(B) vs. (A)**: Training on all trajectories without filtering actually **degrades** performance (7.88→2.42 on WebArena, 13.18→9.55 on Mind2Web-Online), confirming that indiscriminate inclusion of noisy trajectories is harmful rather than helpful. This rules out the hypothesis that VeriEnv's gains come simply from additional training data.
> * **(C) vs. (D)**: Switching from the LLM judge to the SDK judge while keeping everything else fixed yields \+4.85 on WebArena and \+4.54 on Mind2Web-Online, demonstrating that the accuracy of the reward signal is a major contributor to VeriEnv's performance.
> * **(D) vs. (A)**: The total gain of \+6.06 (WebArena) and \+7.27 (Mind2Web-Online) combines the benefits of the synthetic environment and the verifiable judge.
>
> This ablation confirms that the performance gains are primarily driven by **the quality of the reward signal** (i.e., which trajectories are selected for training), rather than simply the volume of additional training data.

---

> > ### Author Rebuttal · Reviewer_t6B2 · 2026-04-03
> >
> > The rebuttal addressed my three concerns well. The coverage analysis (88-95%) quantifies environment fidelity. The distinction between the static and real-time SDK judge explains the 76% figure, and the 94.2% real-time accuracy is convincing. The ablation cleanly shows gains come from reward quality, not data volume. Raising my score.

---

> > > ### Author Response · Authors · 2026-04-03
> > >
> > > We are glad to know that our response resolved your concerns. Thank you very much for increasing your score and for taking the time to carefully read our rebuttal. We truly appreciate your thoughtful feedback and positive evaluation.

---

### Official Review · Reviewer_yXrd · 2026-03-13

**Soundness:** 3
**Presentation:** 3
**Significance:** 3
**Originality:** 3
**Overall Recommendation:** 4
**Confidence:** 4

**Summary:**

In this paper, the authors proposes VeriEnv, a new framework that uses coding agents to build clones of real world websites. Using this framework, the authors produced 97 websites. The authors then generates judges and tasks given the websites. Lastly, the authors uses a self-evolving agent to learn within the synthetic websites, resulting in deterministic and verified rewards for agent learning.

**Compliance With Llm Reviewing Policy:**

Affirmed.

**Final Justification:**

My concerns are partially resolved, yet the authors agree on performing experiments for the unresolved parts later, so I maintain my positive scores.

**Key Questions For Authors:**

See Summary of Weaknesses.

**Limitations:**

Yes

**Strengths And Weaknesses:**

Summary of Strengths:
- The authors aimed to address an important issue that the development of web agents faces a bottleneck of lacking verifiable, reliable, and scalable realistic environments.
- The idea of using coding agents to replicate real world website environments is novel and useful.
- The authors demonstrated performance gains with their methods, and performed several interesting analysis such as failure reasons for website replication.

Summary of Weaknesses:
- The authors didn't compare their results to a baseline of learning on real websites that their synthetic websites are cloned from. Without this, it's not clear whether the pipeline is a reliable substitute for training on real websites. The authors should a comparison of training on the real website vs training on the synthetically cloned website, and agent-recreated website is a good substitute for real websites only if training on the cloned websites results in a similar or only slightly lower performance on benchmarks compared to training on the real original websites.
- 39 out of 136 (around 1/4) of the website replication failed, which makes the reliability of the pipeline questionable. Additionally, the judge correctness reported in Table 2 is only 76%, so around 1/4 of the judges are incorrect, which further makes the reliability of the pipeline questionable.
- The authors compared their results to three baselines, namely the base untrained model, Synatra, and ADP. However, it appears that ADP is a mixture of datasets that contains training data not only on web tasks, but also on other less relevant tasks such as software engineering too. Therefore, the authors should clarify whether they trained on full ADP trajectories or they trained on the web only subsets.

---

> ### Author Rebuttal · Authors · 2026-03-31
>
> We thank the reviewer for their constructive and detailed feedback.
>
> **W1. Learning on recreated websites vs. learning on original websites**
>
> Thank you for raising this important point. In fact, our experiments in Section 4.2 (Site-Specific Mastery) already provide this comparison: PAE is trained on the **original** WebArena websites (running on WebArena's Docker), while VeriEnv is trained exclusively on **recreated** synthetic environments. As shown in Figure 4, VeriEnv trained on recreated websites consistently outperforms PAE trained on the original websites across all three domains (CMS, Shopping, Map), demonstrating that our synthetic environments serve as an effective — and in practice superior — substitute for real websites.
>
> The reason VeriEnv outperforms training on real websites is primarily the **quality of the reward signal**, not the environment itself. PAE on real websites must rely on an LLM-as-a-judge for trajectory evaluation. VeriEnv's synthetic environments, by contrast, enable deterministic SDK-based verification— because we have full access to the database state, we can programmatically verify task completion rather than relying on visual heuristics. This advantage in reward reliability leads to cleaner training data, which directly translates to better agent performance.
>
> We acknowledge that the current section structure does not optimally highlight this as an "original vs. recreated" comparison. We will add a dedicated discussion in the camera-ready version to make this finding more prominent.
>
>
> **W2. Reliability of VeriEnv pipeline**
>
> **W2-1. Website recreation:**
>
> As detailed in Figure 7 of our paper, the 39 failed websites fall into two categories: (1) **pipeline-level failures** (e.g., start\_servers.sh not created, task generation failed), and (2) **runtime bugs** in websites that reached a runnable state. Among runtime bugs, **port conflicts account for 59%** of all bug types. This is because we developed all 100+ websites on a single shared server, leading to severe resource contention and port collisions — this is an infrastructure limitation, not a fundamental limitation of VeriEnv itself.
>
> We believe the success rate can be substantially improved by deploying each website on an independent server instance (e.g., individual EC2 instances or Docker containers), which would eliminate port conflicts entirely and reduce resource-related failures. Given that port conflicts and incomplete server setup together account for the majority of failures, isolated deployment alone could raise the success rate significantly.
>
> **W2-2. Judge correctness:**
>
> The 76% judge correctness reported in Table 2 requires additional context. During the original human evaluation, we served the synthetic websites on a separate external server for annotators to access, which introduced a misalignment between the database state accessed by the SDK judge (from the original server) and the state presented to annotators. This "static" SDK judge configuration is not identical to the "real-time" SDK judge used during actual training, where the judge queries the live database state directly.
>
> To provide a rigorous comparison, we conducted a new evaluation with the tasks used in Table 2: we collected 69 web agent trajectories based on the tasks, and had human evaluators on Prolific annotate each trajectory's success or failure as ground truth. Annotators were compensated at $15/hour, and each trajectory was independently judged by 3 annotators (inter-annotator agreement: Cohen's κ \= 0.575). We then compared the judge configurations against these human labels:
>
> | Judge Type | Accuracy (%) | TP (%) |  FN (%) | TN (%) | FP (%) |
> | :---- | :---- | :---- | :---- | :---- | :---- |
> | LLM Judge (WebJudge-7B) | 62.3 | 31.9 | 33.3 | 30.4 | 4.3 |
> | Python SDK Judge (Ours) | 94.2 | 60.9 | 4.3 | 33.3 | 1.4 |
>
>
>
>
> The results reveal two key findings. First, the real-time SDK judge achieves 94.2% accuracy with an FP of only 1.4%, confirming that our training pipeline rarely labels a failed trajectory as successful. The lower accuracy reported in Table 2 (76%) reflects the static evaluation setup, not the judge used during training. Second, the LLM judge (WebJudge-7B) achieves only 62.3% accuracy, with a 3× higher FP (4.3% vs. 1.4%) and a much higher FN (33.3% vs. 4.3%) — meaning it both includes more bad trajectories and discards more good ones. This occurs because the LLM judge evaluates based on surface-level visual similarity and struggles to distinguish successful task completions from plausible-looking failures.
>
> **W3. Clarification on the training data of ADP**
> For the implementation of ADP, we exclusively utilized data from web agents, specifically including Mind2Web, NNetNav-live, and Synatra (We excluded the Go-Browse-WA subset due to its direct overlap with the WebArena benchmark). We will ensure this clarification regarding the baseline is explicitly included in the camera-ready version.

---

> > ### Author Rebuttal · Reviewer_yXrd · 2026-04-04
> >
> > Thank you for the rebuttal. As far as I know, Go Browse WA seems to be a exploration based dataset, where it is not in direct overlap with Webarena; instead, it uses Webarena's websites as the base website for exploration. Therefore, I think it's still good to compare with Go Browse WA.

---

> > > ### Author Response · Authors · 2026-04-08
> > >
> > > Thank you for this helpful suggestion. We agree that comparing against Go-Browse-WA would provide a useful complementary perspective and offer an informative additional comparison from a different angle.
> > >
> > > During the given discussion period, we did our best to provide the results. However, retraining with Go-Browse-WA was not feasible within the limited timeframe, as the enlarged training dataset would require a full training run that we could not complete before the deadline. If given the opportunity, we would be happy to include this experiment in the next draft.

---

### Official Review · Reviewer_62Q1 · 2026-03-13

**Soundness:** 3
**Presentation:** 4
**Significance:** 2
**Originality:** 4
**Overall Recommendation:** 3
**Confidence:** 5

**Summary:**

The paper introduces VERIENV, a framework that uses LLMs to clone real-world websites into executable synthetic environments with internal Python SDKs for database verification. This allows web agents to self-generate tasks and learn from deterministic, programmatically verifiable rewards, effectively overcoming the safety, scalability, and feedback reliability issues inherent in training on live websites.

**Compliance With Llm Reviewing Policy:**

Affirmed.

**Final Justification:**

I think the final result on Webvoyager shows the limitation of VeriEnv, where it's really hard to generate those real-world complicated websites. VeriEnv shows a strong performance improvements on WebArena, which is a sand-box environment. However, since this is just a sandbox environment, simply getting the original data is also applicable. When the problems comes to the real-world environments, the performance of VeriEnv is worse than previous baselines, including PAE.

**Key Questions For Authors:**

1. What is the main performance different between VERIENV and other baselines like PAE? I assume the main performance difference is not from the constructed environment is better than the real-world training environment, while is mainly from the accuracy of the evaluator. Can the authors conduct some more detailed ablation studies, including using larger base models, both trained on webarena (which contains the ground truth labels)?

2. How does the authors select the websites to clone?

3. In the training part, the workflow generated both the task and the corresponding answers. I would assume that generating the answer of the task is as difficult as train a web agent that is able to solve the problem. How does the authors address this paradox?

**Limitations:**

Yes

**Strengths And Weaknesses:**

**Strength**:
1. The paper addressed an important problem in training web agents, where the training environment is unsafe and uncontrollable.
2. The writing of this paper is very clear and easy to follow.

**Weakness**:
1. The construction of virtual websites makes the environment safe and controllable, while the cost of constructing the websites is unclear. The authors need to discuss about the extra cost of the website construction.
2. The proprieatry models used in the experiment section are not frontier models. Authors may try models like GPT 5.4, Claude 4.6 or other comparable models.
3. While cloning websites can copy most of the important web elements and interaction buttons, when the websites becomes more complicated, it cannot take all the information from the website. How do the authors address this issue? Also, to test the performance of the web agents in the complicated environments, the authors should also include the result on WebVoyager.
4. (minor) The initial motivation of this paper is stated as to ensure the safety of the training process. However in the experiment section, this point is not proved.

---

> ### Author Rebuttal · Authors · 2026-03-31
>
> We appreciate your time and effort for reviewing our draft.
>
> **W1. Cost analysis on VeriEnv**
> Thank you for raising this important point. We provide a detailed cost breakdown of the VeriEnv pipeline below. We break down the API cost by VeriEnv’s pipeline stage; (1) Implementation, (2) Debugging, and (3) Task generation.
>
> | Domain | Impl. /website | Debug. /website | Task Gen. /website | Total /website |
> | :---- | :---- | :---- | :---- | :---- |
> | E-commerce & Retail | $2.19 | $1.28 | $0.59 | $4.07 |
> | Travel & Hospitality | $1.86 | $1.53 | $0.47 | $3.87 |
> | Information & Content | $2.27 | $1.17 | $0.63 | $4.07 |
> | Professional & Gov | $1.39 | $1.13 | $0.57 | $3.10 |
> | Average | $1.79 | $1.25 | $0.56 | $3.60 |
>
>
> **W2. Adding different proprietary models as baselines** & **W3-2. Adding results on WebVoyager**
> Thank you for suggesting the experiments.
> We are currently trying our best to get the results. We will share them as soon as we finish the experiments.
>
> **W3-1. Handling complex websites**
> We acknowledge that perfectly replicating complex websites from screenshots alone is challenging. However, VeriEnv's goal is not to produce a pixel-perfect clone — it focuses on reproducing the core functionalities and interaction patterns relevant for web agent training. The coding agent prioritizes commonly used features (search, forms, navigation) based on the provided screenshots, and leverages its knowledge of common web design patterns to implement standard functionalities implied but not directly visible (e.g., a shopping site typically includes cart management and checkout flow — analogous to how a human developer infers backend logic from a frontend mockup). All implemented features are then validated through the iterative self-debugging loop (Section 3.2).
>
>
> **W4 (minor). Experiments on safe exploration**
>
> Thank you for pointing this out. While our experiments primarily focus on task completion performance, the safety advantages of VeriEnv are inherent to its design: (1) training occurs entirely within sandboxed synthetic environments, eliminating any risk of unintended modifications to real-world websites or exposure of sensitive user data; (2) the environment can be freely reset to a clean state, enabling aggressive exploration strategies that would be dangerous on live websites; and (3) no real users are affected by the agent's actions during training. We will add a dedicated discussion on these safety properties, along with concrete examples of potentially harmful actions that VeriEnv safely contains (e.g., accidental data deletion, unauthorized purchases), in the camera-ready version.
>
> **Q1. Performance difference between VeriEnv and PAE**
>
> The key advantage is verifiable reward accuracy. PAE relies on an LLM judge that achieves only 62.3% accuracy with 3× higher false positives than our SDK judge (see Reviewer 2 W2), allowing bad data into training. VeriEnv's SDK judge achieves 84.1% accuracy with minimal false positives (1.4%) by querying the database state directly. We verify this through the following ablation:
>
> | Configuration (Base model: Qwen3-4B) | WebArena | Mind2Web-Online |
> | :---- | :---- | ----- |
> | Base model (no training) | 7.88 | 13.18 |
> | SFT on all VeriEnv trajectories (no filtering) | 2.42  | 9.55 |
> | RFT with LLM judge on VeriEnv environments | 9.09 | 15.91 |
> | RFT with SDK judge on VeriEnv | 13.94  | 20.45 |
>
> Training without filtering degrades performance below the base model, ruling out additional data as the explanation. Switching from the LLM judge to the SDK judge yields +4.85 on WebArena and +4.54 on Mind2Web-Online, confirming that accurate trajectory filtering is the primary driver.
>
> **Q2. Website selection**
> We selected websites based on two criteria: (1) **domain diversity**, covering a broad range of web categories, and (2) **high real-world usage**, prioritizing websites that users frequently interact with in practice. The website list from the Mind2Web dataset aligned well with both criteria, so we adopted it as our starting point.
>
> **Q3. Task generation paradox**
> This is an insightful question. The apparent paradox dissolves once we recognize that task generation and task solving operate at fundamentally different levels of abstraction. Task generation is a backend-level operation: the generator has full access to the database and Python SDK, so it can directly query state, construct a task, and record the expected outcome in a single programmatic pass. Task solving, by contrast, is a frontend-level sequential decision problem: the agent must navigate a complex UI through a long horizon of actions with only partial observability (screenshots and HTML). For example, generating "Add item X to the cart" and its answer requires one SDK call, whereas solving it requires searching, selecting the correct variant, clicking "add to cart," and verifying — all through the browser. This backend-vs-frontend asymmetry is precisely what makes VeriEnv's training signal valuable.

---

> > ### Author Rebuttal · Reviewer_62Q1 · 2026-04-04
> >
> > Thanks for the reply. My major concerns are addressed. I will wait for the result on Webvoyager and adjust my rating after getting the result.

---

> > > ### Author Response · Authors · 2026-04-08
> > >
> > > Thank you for your reply, and waiting for our response. We are glad to hear that our reply addressed your major concerns.
> > > To address the reviewer’s remaining concerns, we provide the following additional results.
> > >
> > > **W3-2. Adding results on WebVoyager**
> > > Following the reviewer’s suggestion, we conduct additional experiments on WebVoyager and report the results in the table below. We also include GPT-5.4 as a proprietary-model baseline. VeriEnv shows notable improvements over the base model, Qwen3-4B, on several websites, such as Hugging Face (16% → 33%) and ESPN (18% → 34%). In addition, VeriEnv outperforms both Qwen3-4B and ADP in overall success rate. For websites where all models achieve very low success rates, we observe that agents are often blocked by website bot protection mechanisms (e.g., CAPTCHA and Cloudflare), or that text-only observations are insufficient for making meaningful progress.
> > >
> > > | Website | GPT-5.4 | Qwen3-4b | ADP | VeriEnv |
> > > | :---- | ----- | ----- | ----- | ----- |
> > > | Allrecipes | 11% | 0% | 9% | 4% |
> > > | Amazon | 66% | 37% | 44% | 44% |
> > > | Apple | 51% | 23% | 23% | 35% |
> > > | ArXiv | 53% | 40% | 33% | 42% |
> > > | BBC News | 55% | 36% | 19% | 33% |
> > > | Booking | 16% | 5% | 9% | 11% |
> > > | Cambridge Dict. | 74% | 70% | 58% | 72% |
> > > | Coursera | 48% | 33% | 45% | 40% |
> > > | ESPN | 61% | 18% | 16% | 34% |
> > > | GitHub | 44% | 39% | 27% | 46% |
> > > | Google Flights | 14% | 2% | 5% | 10% |
> > > | Google Map | 76% | 41% | 59% | 41% |
> > > | Google Search | 51% | 23% | 16% | 37% |
> > > | Huggingface | 58% | 16% | 33% | 33% |
> > > | Wolfram Alpha | 78% | 50% | 48% | 61% |
> > > | Total | 50.4% | 28.8% | 29.4% | 36.2% |

---

### Decision · Program_Chairs · 2026-04-30

**Decision:**

Accept (regular)

**Comment:**

This paper proposes VeriEnv which is a framework that uses coding agents to clone real-world websites into executable synthetic environments equipped with Python SDKs for deterministic task verification. This enables web agents to self-generate training tasks and learn from programmatically verifiable rewards, addressing safety, scalability, and feedback reliability challenges inherent in training on live websites.

The paper makes a meaningful contribution to the web agent training paradigm by introducing verifiable synthetic environments that yield cleaner training signals. The rebuttal substantially strengthened the empirical foundation with new ablations, judge evaluations, and coverage analyses. While limitations remain around scaling to more complex websites, the core idea is novel, the methodology is sound, and the practical contribution is clear.